# Inundation analysis of metro systems with SWMM incorporated in GIS: a case study in Shanghai

Hai-Min Lyu[1], Shui-Long Shen[2,3], Jun Yang[4], Zhen-Yu Yin[5]

[1]Department of Civil Engineering, School of Naval Architecture, Ocean, and Civil Engineering, Shanghai Jiao Tong University, Shanghai 200240, China

[2]Key Laboratory of Intelligent Manufacturing Technology (Shantou University), Ministry of Education, Shantou, Guangdong 515063, China

[3]Department of Civil and Environmental Engineering, College of Engineering, Shantou University, Shantou, Guangdong 515063, China

[4]Department of Civil Engineering, The University of Hong Kong, Pokfulam, Hong Kong, China

[5]Department of Civil and Environmental Engineering, The Hong Kong Polytechnic University, Hung Hom, Kowloon, Hong Kong, China

*Correspondence to*: Shui-Long Shen (shensl@stu.edu.cn)

**Abstract**. This study presents an integrated approach to evaluate inundation risks, in which an algorithm is proposed to integrate the storm water management model (SWMM) into geographical information system (GIS). The proposed algorithm simulates the flood inundation of overland flows and in metro stations for each designed scenario. It involves the following stages: i) determination of the grid location and spreading coefficient, and ii) an iterative calculation of the spreading process. In addition, an equation is proposed to calculate the inundation around a metro station and to predict the potential inundation risks of the metro system. The proposed method is applied to simulate the inundation risk of the metro system in the urban centre of Shanghai under 50-year, 100-year, and 500-year rainfall intensities. Both inundation extent and depth are obtained and the proposed method is validated with records of historical floods. The results demonstrate that in the case of a 500-year rainfall intensity, the inundated area with a water depth excess of 300 mm covers up to 5.16 km$^2$. In addition, four metro stations are inundated to a depth of over 300 mm.

**Keywords:** urban inundation, scenario analysis, metro system, algorithm for overland flow, SWMM, GIS

# 1. Introduction

With rapid urbanisation, numerous urban constructions (such as underground metro systems, malls, infrastructural systems, and parks) have been built (Wu et al., 2016, 2019; Peng and Peng, 2018). Underground constructions (Shen et al., 2014, 2017; Tan et al., 2017; Wang et al., 2019) render the environment susceptible to certain natural hazards, such as floods, tornados, and typhoons (Lyu et al., 2016, 2017). Recently, climate change has resulted in various rainstorm events in China (Zhou et al., 2012; Yin et al., 2017, 2018; Xu et al., 2018). Many metropolitan areas have frequently suffered from inundation owing to urban flooding, which is one of the most severe hazards and causes the catastrophic submerging of ground surfaces and severe inundation of underground facilities. Numerous metro lines were inundated during the flood season (May–September) in 2016 in China; for instance, the metro lines in Guangzhou and Wuhan. The Shanghai Station of the metro line No. 1 was inundated on 3 October, 2016 (Lyu et al., 2018a, b, 2019a, b, c). Thus, the prediction and prevention of inundation in metro systems must be integrated into current urban layouts (Huong and Pathirana, 2013).

In general, four methods have been developed to predict the inundation risk: (1) a statistical analysis based on historical disaster records (Nott, 2006), (2) geographical information system and remote sensing (GIS–RS) techniques (Sampson et al., 2012; Meesuk et al., 2015), (3) multi-criteria index analysis (Jiang et al., 2009; Kazakis, 2015), and (4) scenario-based inundation analysis (Willems, 2013; Naulin et al., 2013). Although assessment results based on historical disaster records can predict the inundation risk of an area, the method requires high numbers of data (Nott, 2006). The GIS–RS method can provide the technological support for an inundation risk evaluation (Sampson et al., 2012; Meesuk et al., 2015). However, it requires high investments and high-resolution data sources. The multi-criteria index analysis has a few limitations regarding the determination of subjective indices (Jiang et al., 2009; Kazakis, 2015). The scenario-based inundation analysis predicts the inundation risk for different scenarios (Willems, 2013;

Naulin et al., 2013; Wu et al., 2018) and requires the topography, land use, and urban drainage system data. Owing to the complex interaction between the drainage system and overland surface in urban regions, scenario-based models can only simulate inundation for a small range (e.g. below 3 km$^2$) (Wu et al., 2017), which limits their application. Thus, scenario-based models must be extended to apply them to large-scale overland flow problems (e.g. entire regions with areas of over several hundred square kilometres).

Numerical simulation is a useful tool to analyse urban flooding. Xia et al. (2011) integrated an algorithm into a two-dimensional (2D) hydrodynamic model to assess flood risks. Szydlowski et al. (2013) proposed a numerical flood model, in which a mathematical model was incorporated into a 2D hydrological model to estimate inundation risks. Furthermore, Chen et al. (2015) used numerical simulations to predict the inundation risk in a flood-prone coastal zone. Morales-Hernandez et al. (2016) presented coupled one-dimensional (1D) and 2D models (1D–2D models) for the fast computation of large-river floods. However, these numerical models have the following deficiencies: i) the characteristics of the landform (e.g. the topographical elevation, slope, and river system) are difficult to model; ii) a numerical simulation is typically used to estimate the inundation risk in a small area, whereas flooding hazards often occur on a regional scale. The existing methods can only simulate inundation for small ranges (e.g. several square kilometre) (Naulin et al., 2013; Wu et al., 2018). Therefore, a new tool (e.g., the GIS technique) is required to consider variations in topographical elevations. Moreover, an integrated method is required to simulate regional-scale flooding.

The storm water management model (SWMM) is a dynamic hydrological model, which is widely used to simulate the rainfall runoff processes in urban catchments (Shen and Zhang, 2015; Bisht et al., 2016; Ai–Mashaqbeh and Shorman, 2019; Zhao et al., 2019). However, the SWMM cannot be used to determine surface water flows directly. The existing researches have been applied to only small regions of several square kilometres (Wu et al., 2018; Chen et al., 2018; Kumar et al., 2019). For example, Zhu et al., (2016)

applied the SWMM and a multi-index system to evaluate the inundation risks in southwest Guangzhou, China (area of 0.43 km$^2$). Feng et al. (2016) selected the SWMM as modelling platform to simulate the inundation risks for a campus in Salt Lake City, Utah, US (area of 0.11 km$^2$). Moreover, Wu et al. (2017) applied the SWMM in combination with LISFLOOD-FP to simulate urban inundations in Dongguan City, China (area of 2 km$^2$). Predicting the potential inundation risks on a regional scale with the SWMM is challenging, because the determination of the spreading process and flow direction of the runoff for a large scale is difficult. Thus, a new method that can predict the inundation risk on a regional scale with SWMM is required.

Few researchers have focused on the inundation risks of metro systems. Yanai (2000) and Hashimoto (2013) analysed the flood event in Fukuoka City in 1999, which led to a serious inundation of the metro station. Based on previous research, Aoki (2016) proposed anti-inundation measures for the Tokyo Metro. Herath and Dutta (2004) created an urban flooding model and included the underground space. Furthermore, Suarez et al. (2005) conducted a flooding risk assessment for the Boston metro area. Ishigaki et al. (2009) presented a method for the safety assessment of a Japanese metro. Nevertheless, research and literature on the inundation risks of metro systems are insufficient.

The objectives of this study are as follows: i) propose a method for the prediction of the potential inundation risk on a regional scale with a new algorithm that integrates SWMM into a GIS for overland flow simulations; ii) propose an evaluation method for the potential inundation risk of a metro system. Then, the proposed method is adopted to simulate urban inundations and inundation depths for the Shanghai Metro system under 50-year, 100-year, and 500-year-rainstorm events. The proposed method assumes that the runoff on the surface flows from one subcatchment to another.

## 2. Materials

### 2.1 Study area

Shanghai is located at 31°20′–31°00′N (latitude) and 121°20′–121°31′E (longitude) and covers more than 6340 km$^2$. Fig. 1 shows the administrative region and metro line distribution of Shanghai. The Shanghai Metropolis is surrounded by the Yangtze River in the Northeast, Hangzhou Bay in the Southeast, Zhejiang Province in the West, and Jiangsu Province in the Northwest (Shen and Xu, 2011). The average elevation ranges from 2 to 5 m above the sea level in Shanghai (Xu et al., 2019). The urban centre (area of 120 km$^2$) includes the districts of Jingan, Huangpu, Luwan, Xunhui, Changning, Putuo, Zhabei, Hongkou, and Yangpu. Metro line No. 1 was constructed in Shanghai between 1990 and 1995. The first metro line (Xujiahui Station to Jinjiang Park Station) was opened for operation on 28 May 1993. Currently, 14 metro lines are under operation (data originating from Planning of Shanghai Metro Line, 2016), and 8 metro lines are under construction. As shown in Fig. 1, the urban centre has a dense distribution of metro lines and is located near the Huangpu River, which passes through Shanghai City. Several metro lines pass under the Huangpu River. The rising tide in the Huangpu River increases the flood risk; particularly during the flood season. As significant underground infrastructures, metro lines play important roles at traffic junctions in mega-cities. During flood disasters, metro lines become crippled, which results in severe consequences such as a traffic paralysis.

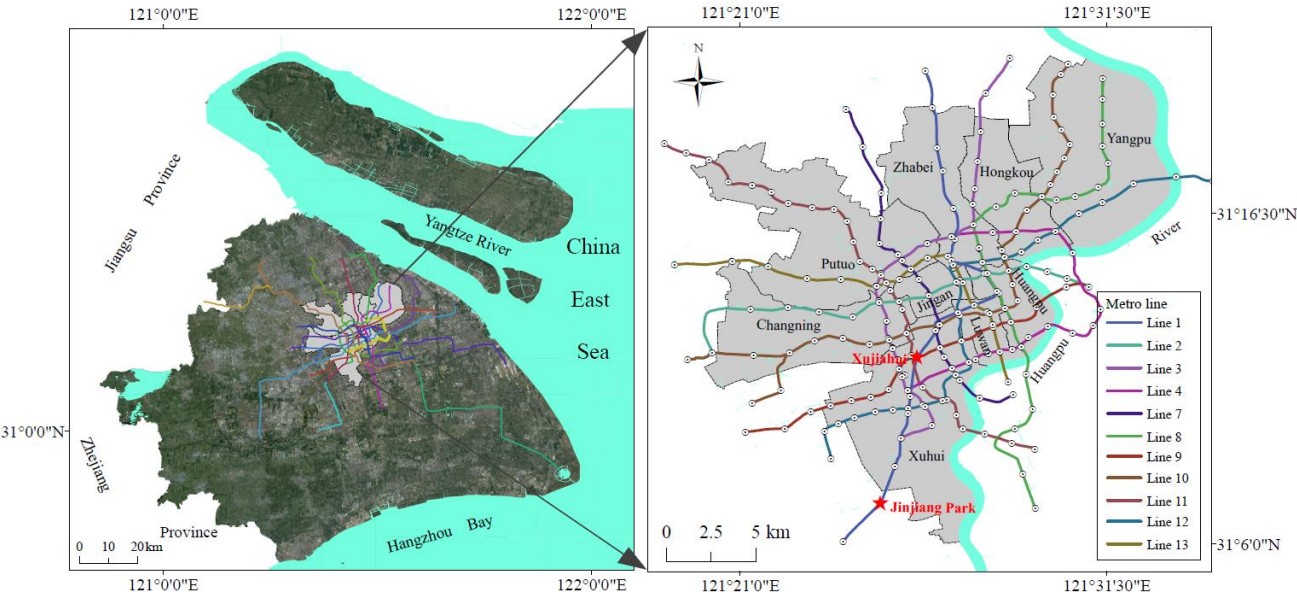

**Figure 1:** Metro line distribution in the study area of Shanghai

## 2.2 Precipitation data and processing

Precipitation is the external driving force behind flood disasters. The Chicago design storm (Yin et al., 2016a, b) is widely applied to generate rainfall hyetographs, which are used to calibrate the peak intensity and precipitation before and after the peak for different return periods of the rainfall. The equations for the Chicago design storm are as follows:

$$i_a = \frac{a \times [\frac{(1-c) \times t_a}{1-r} + b]}{(\frac{t_a}{1-r} + b)^{c+1}}, \tag{1}$$

$$i_b = \frac{a \times [\frac{(1-c) \times t_b}{r} + b]}{(\frac{t_b}{r} + b)^{c+1}}, \tag{2}$$

where $i_a$ and $i_b$ are the precipitation intensities after and before the peak value (mm/min), respectively; $t_a$ and $t_b$ are the times after and before the peak value (min), respectively; $a$, $b$, and $c$ are specific values related to the local municipal rainstorm models of the intensity–duration–frequency (IDF) type.

Based on investigations, the IDF of the Shanghai municipal rainstorm can be expressed as follows (Jiang et al., 2015; Yin et al., 2016a):

$$i = \frac{9.581(1+0.8461\lg T)}{(t+70)^{0.656}},$$ (3)

where $i$ is the precipitation intensity (mm/min), $T$ is the return period of the precipitation (year), and $t$ is the precipitation duration (min).

To consider temporal variations, the parameter $r$ (the ratio of the time necessary to reach the peak to the total event duration) was empirically fixed to 0.45 (Yin et al., 2016a). The parameter $r$ is used to determine the location of the rainfall peak during a rainfall scenario. A rainfall intensity for a duration of 2 h and return periods of 50-year, 100-year, and 500-year scenarios were designed to model the probable inundations.

## 2.3 Topographical data and drainage station

The digital elevation model (DEM) with a 30 m resolution was obtained from a geospatial data cloud. To simulate the reality of the study area, the DEM was further modified with buildings based on field surveys and documents (Yin et al., 2016a). The building heights were rebuilt in the DEM to reproduce the blockage effects of surface flows. During the reprocess of elevation data, the investigated area was divided into grids with 20 m × 20 m in GIS. Since the original elevation of each grid was extracted from the DEM data with spatial resolution of 30 m, the linear interpolation was conducted to convert it into 20 m size. Shanghai is a flat region with an elevation difference of only 2 to 3 m within a range of 25 km. Thus, the accuracy of linear interpolation is enough. Each grid was provided with building distribution data and a DEM. The grids with the original elevation were modified to include the building heights. Because the locations with buildings are not inundated, the modification is reasonable. Since the study area is classified into grids with 20 m × 20 m, the obtained distribution of water depth is within the area of 400

$m^2$. Therefore, the proposed method can achieve an accuracy of the inundation distribution within 400 $m^2$. The distribution of the drainage stations was obtained from the Shanghai Municipal Sewerage Management Branch (Quan et al., 2011).

## 3. Methodology

The SWMM model was incorporated into the GIS model to predict the inundation depth. The following phases must be performed during its incorporation:

(1) The investigated area was classified into different subcatchments in the GIS. Each subcatchment was provided with the corresponding geographical information (e.g. elevation, slope, area, and width). The information of each subcatchment was stored in the GIS database.

(2) The information of each subcatchment was exported from the GIS database and reproduced to produce a '*.inp*' document.

(3) The '*.inp*' document was integrated into the SWMM model to calculate the water volume of each subcatchment.

(4) The calculated water volume of each subcatchment was converted into the average water depth with the water volume and area of each subcatchment.

(5) Each subcatchment was divided into 20 m × 20 m grids in GIS. The study area includes 113810 grids. The information of each subcatchment and average water depth were extracted into the grids in GIS database.

(6) The grids with all information were applied to perform the spreading process with the proposed algorithm in GIS until the water level of each grid was stable. During spreading process, a spreading coefficient was used to move the runoff between neighbouring grids.

Finally, the water depth of each grid was exported to visualise the distribution of the inundation depth of the investigated area. The details of the proposed algorithm are presented in Section 3.2.

## 3.1 SWMM calibration

The SWMM model is widely applied to simulate the runoff quantity produced in each subcatchment in a simulated period. The results obtained with the SWMM model approximate the measured value, and indicate that the runoff reaches a peak in the shortest time possible (Lee et al., 2010). In addition, researchers have reported that the SWMM is one of the best hydrologic models (Tan et al., 2008; Cherqui et al., 2015). It is assumed that under extreme rainfall scenarios, the runoff concentrates at the outlet of each catchment, and the function of the drainage network is negligible. In this case, the overland flow is more likely to move in multiple directions rather than through the predefined flow paths and outlets. Therefore, a coefficient was included in the spreading process algorithm to determine the flow paths on the surface. The spreading coefficient was used to move the runoff between neighbouring subcatchments. Moreover, the function of the drainage network was reflected by the drainage capacity of each drainage station (see Fig. 2). The water quantity of each subcatchment calculated in SWMM was reduced by the capacity of the drainage station. Detailed information on the algorithm is introduced in Section 3.2.

### 3.1.1 Subcatchment division and flow direction

A subcatchment is the basic calculation cell in the SWMM. The two types of subcatchment divisions are based on (Shen and Zhang, 2015): i) the subcatchment partition and ii) drainage system. In this study, a subcatchment was initially divided with the Thiessen polygon method based on the spatial distribution of the drainage stations (Shen and Zhang, 2015; Zhu et al., 2016). The drainage capacity of each subcatchment was determined with the service range of the drainage stations. In addition, the boundary was a fixed boundary. Thus, the water level at the boundary was not considered to spread because the water volume of the grids located at the boundary have less effect on the spreading process. Fig. 2 shows the characteristics of subcatchments and grids in SWMM and GIS. The study area was classified into different subcatchments based on the drainage capacity of the drainage station (Fig. 2a). The drainage

capacity of each drainage station was obtained from the existing publication (Yin et al., 2016b). Each subcatchment was meshed into grids with 20 m × 20 m (Fig. 2b) and included the corresponding information. To realistically mimic the effect of the natural hydrology features of a subcatchment, the topographical characteristics of the catchments were considered during the subcatchment division. The flow direction of each subcatchment was determined based on the DEM. Furthermore, the average elevation and slope of each subcatchment was extracted with the GIS tools. To reproduce obstacles caused by buildings in the surface runoff flow, the elevation included building heights and locations. Thus, the average slope of a subcatchment reflected the obstacle imposed by a building in a rainwater flow.

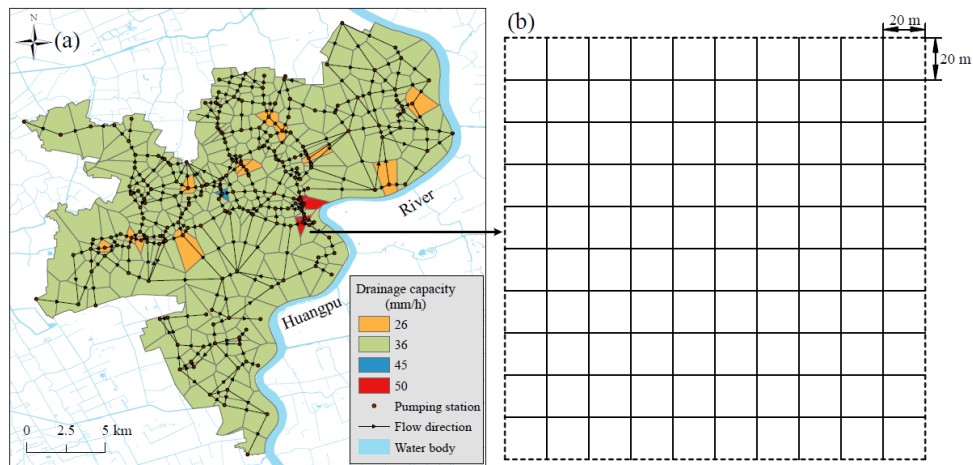

**Figure 2:** Calculated subcatchments and grids in SWMM and GIS: (a) drainage capacity and flow direction of each subcatchment; (b) calculated grid of each subcatchment

### 3.1.2 Model input and determination of parameters

Based on the previously mentioned subcatchment division, each subcatchment was assigned its topographical characteristics. The model included 195 subcatchments and 204 junctions. Each subcatchment in the SWMM model included the width, area, and permeability. The width and area can be calculated with the Spatial Analyst Tools in GIS. Table 1 lists the parameters of the subcatchments in the SWMM. The impervious parameter was determined according to the land use types. The study area

is located in the urban centre, where the land use has no big changes. The dense distribution of buildings leads to an impervious surface of more than 80% of the total surface. Owing to the existence of road pavements, subgrades, and many municipal pipelines under the roads, water infiltration through the road and subsurface is very low. Thus, roads can be considered impervious, and soil infiltration and evapotranspiration have slight effects on the surface runoff concentration during short-term flash flooding during rainstorms. The soil infiltration mainly depends on green land (e.g. lawns, flower beds, and groves) and in the water bodies within the study area. The geotechnical information in Shanghai is as follows: The groundwater table is higher than 2 m below the ground surface. The soil type at a depth of 2 m is mixed soil with sand (5%), silt (55%), and clay (40%) according to the Shanghai Geotechnical Investigation Code (DGJ08-37-2012). The sand content is 15% at the surface. Thus, the soil has a hydraulic conductivity of $2 \times 10^{-5}$ m/s, which is 72 mm/h. At the bottom of the water body, the soil has more clay (>50%) and less sand (<5%) with a hydraulic conductivity of $2 \times 10^{-7}$ m/s, which is 0.72 mm/h (Shen et al., 2015). Based on the SWMM handbook, the maximal infiltration rate was set to 72 mm/h to reflect the characteristics of the green land, whereas the minimal value (0.72 mm/h) reflects the characteristics of the water body because the underlying soil is saturated clay. In addition, the blocking effect of the buildings has a significant influence on the surface runoff generation and concentration. Therefore, the heights of existing buildings were extracted to modify the elevation of the calculated grids.

**Table 1** Parameters of the subcatchments in the SWMM

| Parameter | Meaning | Value |
| --- | --- | --- |
| Area (km$^2$) | Area of each subcatchment | 10.38–0.16 |
| Width (m) | Width of each subcatchment | 5283.83–432.45 |
| Impervious (%) | Percentage of the impervious area | 65–80 |
| Slope (°) | Average slope of each subcatchment | 0.3–5.5 |
| Destore-impervious (mm) | Depression storage depth in the impervious area | 1.5 |
| Destore-pervious (mm) | Depression storage depth in the previous area | 5 |
| N-impervious | Manning's coefficient in the impervious area | 0.1 |
| N-pervious | Manning's coefficient in the previous area | 0.24 |

| MaxRate (mm/h) | Maximum infiltration rate | 72 |
| MinRate (mm/h) | Minimum infiltration rate | 0.72 |
| Decay (h$^{-1}$) | Decay constant | 4 |
| Dry (d) | Drying time | 2 |

## 3.2 Spreading algorithm

After the calibration of the runoff volume of each subcatchment, the spreading procedure of the calibrated runoff must be performed. The spreading procedure algorithm was used to exchange the data between GIS and SWMM. Fig. 3 presents the spreading procedure of the runoff. Furthermore, Fig. 3(a) illustrates the determination of the grid location and spreading coefficient. Fig 3(b) presents an iterative calculation of the spreading process. First, the study area was meshed into 113810 grids with 20 m × 20 m with the GIS fishnet tools [see Fig. 3(a)]; second, the calculated average inundation depth was extracted from each grid in GIS [see Fig. 3(b)]. As shown in Fig. 3(b), the spreading procedure includes four steps:

*Step 1*: The grid location (*GL*) and spreading coefficient (*f*) are determined (see Fig. 3a). Each grid $h_I$ is surrounded by $h_{Ij}$ grids ($j$ = 1, 2,…, 8). If the grid is surrounded by eight grids ($h_I + \Delta x = h_{Ij}$ or $h_I + \Delta y = h_{Ij}$), the locations of grids $h_{Ij}$ are determined with *GL* = 1 and spreading coefficient $f$ = 1. However, if the grid is located at a boundary and surrounded by fewer than eight grids ($h_I + \Delta x = h_{Ij}$ and $h_I + \Delta y = h_{Ij}$), the locations of grids $h_{Ij}$ are determined with *GL* = -1 and spreading coefficient $f$ = 0.569.

*Step 2*: The spreading grid is classified into a rank. In this process step, the rank of a spreading grid is based on the possible water quantity in target grid $h_I$ coming from the surrounding grids $h_{Ij}$ and can be described with Eq. (4). It is assumed that the grid with maximal water quantity is the first spreading grid.

$$Q_{t \arg et} = \sum_{j=1}^{n} h_{Ij} \cdot a_j \cdot f \quad (n = 1, 2, \cdots, 8),$$ (4)

in which $Q_{t \arg et}$ is the runoff of target grid $j$; $h_{Ij}$ implies that $j$ grids surround grid $h_I$; $a_j$ is the area of grid

*j* (in this study, $a_j = 400\ \text{m}^2$) and *f* is the spreading coefficient.

*Step 3*: The spreading and updating of the water level in each grid starts. The water level difference in each spreading step is assumed to be $\Delta h$ ($\Delta h$ can be fixed to a specific or flexible value). The water quantity in each spreading step can be described with Eq. (5). To ensure the convergence of the calculation process, the value of $\Delta h$ can be fixed to a small value in the initial stage, and then increased with time. In this study, the initial $\Delta h$ was fixed to 0.01.

$$Q_{spreading} = \sum_{j=1}^{n}(h_I - h_{Ij})\cdot a_j \quad (n=1,2,\cdots,8), \tag{5}$$

where $Q_{spreading}$ is the runoff of the surrounding grids.

*Step 4*: The spreading cessation is estimated. When the water level difference between target grid $h_I$ and surrounding grid $h_{Ij}$ is below 0.01, the spreading process is stopped. The pseudo-code for this algorithm is provided in the Appendix.

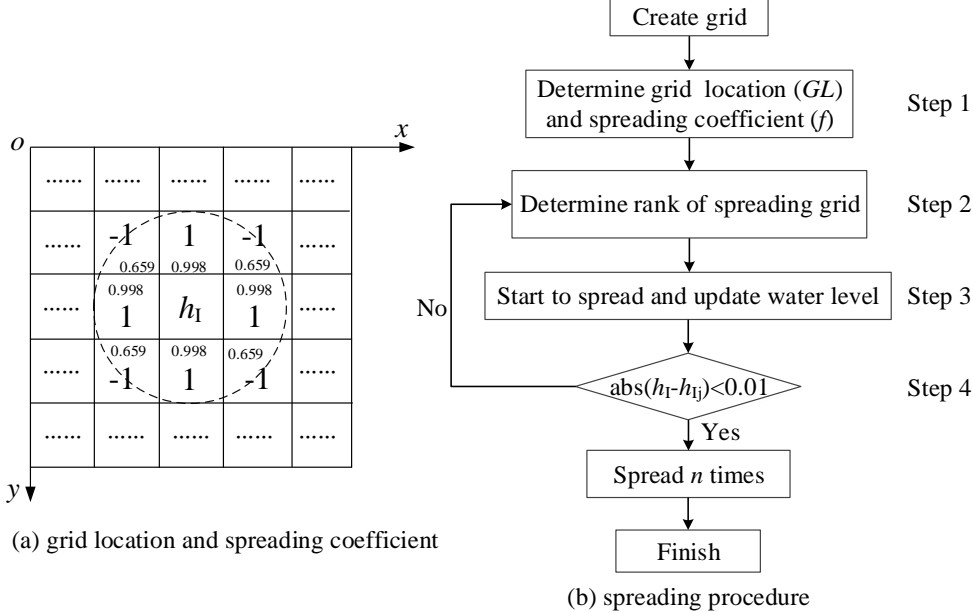

(a) grid location and spreading coefficient

(b) spreading procedure

**Figure 3:** Description of the spreading process: (a) determination of the grid location and spreading coefficient and (b) iterative calculation of the spreading process

The inundation depth around a metro station is used to evaluate the inundation risk of metro lines. Therefore, Eq. (6) is proposed:

$$h_{t(station)} = h_i - \frac{p}{A} - h_{0(station)} \, ,$$ (6)

where $h_{t(station)}$ is the inundation depth around the metro station, $h_i$ the inundation depth on the ground surface after flooding events, $p$ the drainage capacity of the metro station, $A$ the inundation area, and $h_{0(station)}$ the step height of the metro station (based on the design standards of metro systems, $h_{0(station)} = 0.2$ m). According to the code for metro design (GB 50157-2013), the drainage capacity of a metro station is determined according to the local IDF with a duration of 5 min for a 50-year rainfall intensity. In this study, the drainage capacity of the metro station was determined with Eq. (3) for a 50-year rainfall intensity and duration of 5 min. When $h_{t(station)} > 0$, the metro station becomes inundated.

## 3.3 Model calibration and visualisation

During the establishment of the storm water model in the SWMM, the rainfall intensity was set to return periods of 50-year, 100-year, and 500-year. In addition, the rainfall duration was set to 2 h. The water volume of each subcatchment can be computed in the SWMM. The calculated water volume in the SWMM model was input into the GIS model to update the water level of each grid with the proposed algorithm. The stable water level of each grid in GIS was used to reflect the inundation depth in the study area. Subsequently, the inundation depth was used to evaluate the flood risks. By using the inundation depth of the ground surface, the inundation depth around a metro station can be determined with Eq. (6). Furthermore, the spatial distribution of the inundation depth can be visualised with the GIS. The calibration of the proposed model is based on a comparison between the predicted results and historic inundation locations.

## 4. Results and analysis

### 4.1 Runoff volume

Fig. 4 shows the runoff volume of each subcatchment at different rainfall intensities. The runoff increases with increasing rainfall intensity. Furthermore, the area of each subcatchment is presented. Most subcatchments cover approximately 2 km$^2$. The area of each subcatchment was used to calculate the average water depth. Then, the average water depth was used to simulate the spreading process with the proposed algorithm. The computed SWMM results are incorporated into GIS to obtain a map of the inundation distribution. Because the average water depth is related to the runoff volume and area of a subcatchment, the simulated results reflect the surface runoff and overland flow of the study area. Thus, the algorithm plays an important role in the analysis of the spreading processes of surface runoff volumes.

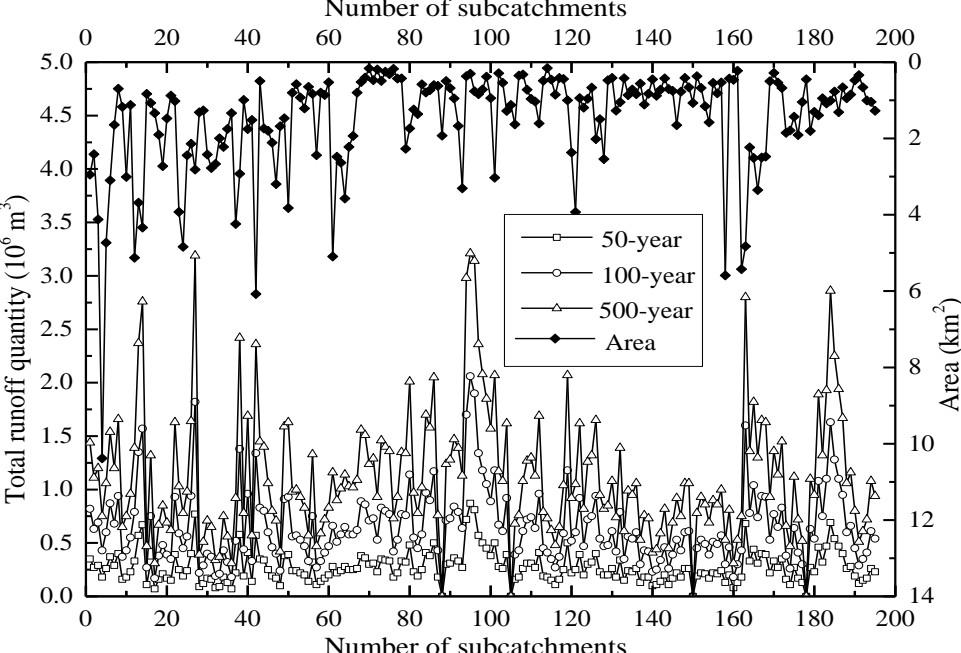

**Figure 4:** Runoff volume of each subcatchment in the corresponding area under different rainfall intensities

## 4.2 Inundation extent and depth

The inundation depth across the study area was computed with the proposed algorithm (see Fig. 3). Fig. 5 displays the distribution of the inundation extent and depth for the previously mentioned rainfall scenarios. The floodwater profiles of the three scenarios are similar. However, an increasing rainfall intensity exacerbates the inundation depth and areal extent. Figs. 5a and 5b exhibit similarities in inundation depths and extents for the different scenarios. Fig. 5c presents the maximal inundation depth and extent for the 500-year rainfall intensity. The maximal depth for each scenario first occurs in certain places in the Changning, Huangpu, and Yangpu districts. The maximal inundation depth exceeds 400 mm.

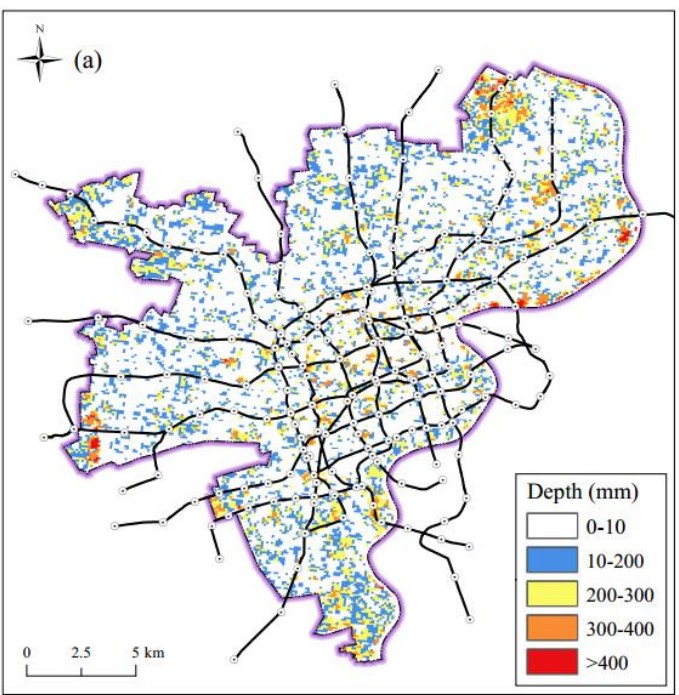

**Figure 5:** Distribution of the potential inundation extent and depth under different rainfall intensity: (a) 50-year, (b) 100-year, and (c) 500-year-rainfall intensity (continuing)

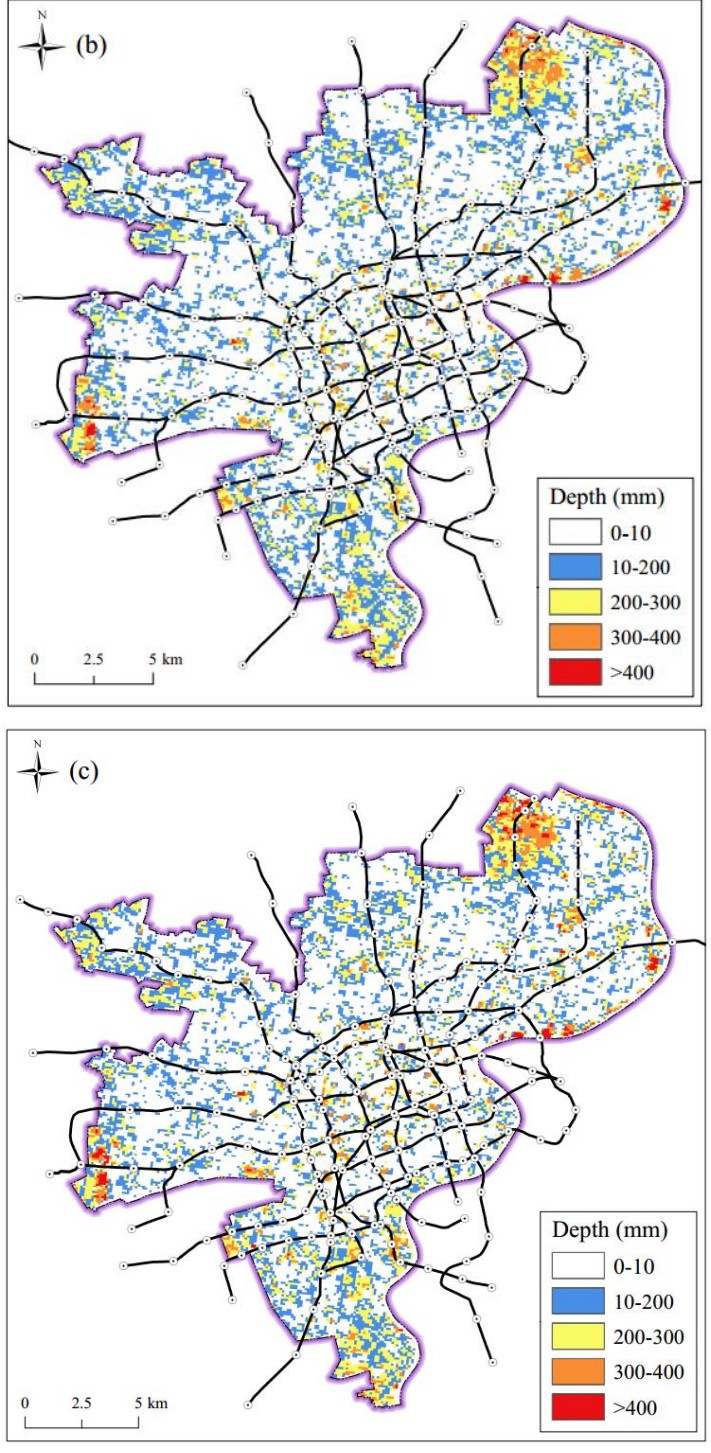

**Figure 5:** Distribution of the potential inundation extent and depth under different rainfall scenarios: (a) 50-year, (b) 100-year, and (c) 500-year-rainfall intensity (continued)

To analyse the inundation risks of different scenarios, the inundated area and ratio were determined with GIS. The inundation ratio is represented by the ratio of the inundated area to the total area (120 km$^2$). Fig. 6 presents the inundated area and ratio for different inundation depths. The inundation depth of 300 mm constitutes a key point in the variation patterns of the three scenarios; specifically, when the inundation depth is over 300 mm, the inundated area increases with increasing rainfall intensity. When the inundation depth is below 300 mm, the variations in the inundated area do not exhibit this pattern. To illustrate the variations in the inundated area and ratio for an inundation depth of over 300 mm, their values for a depth range of 300–400 mm and above are presented in Fig. 6. When the inundation depth is below 300 mm, the inundated area exhibits an irregular distribution pattern, which might be due to the landform. The inundation area for an inundation depth of over 300 mm is up to 5.16 km$^2$ for a 500-year rainfall intensity, which is 4.3% of the total studied area (120 km$^2$).

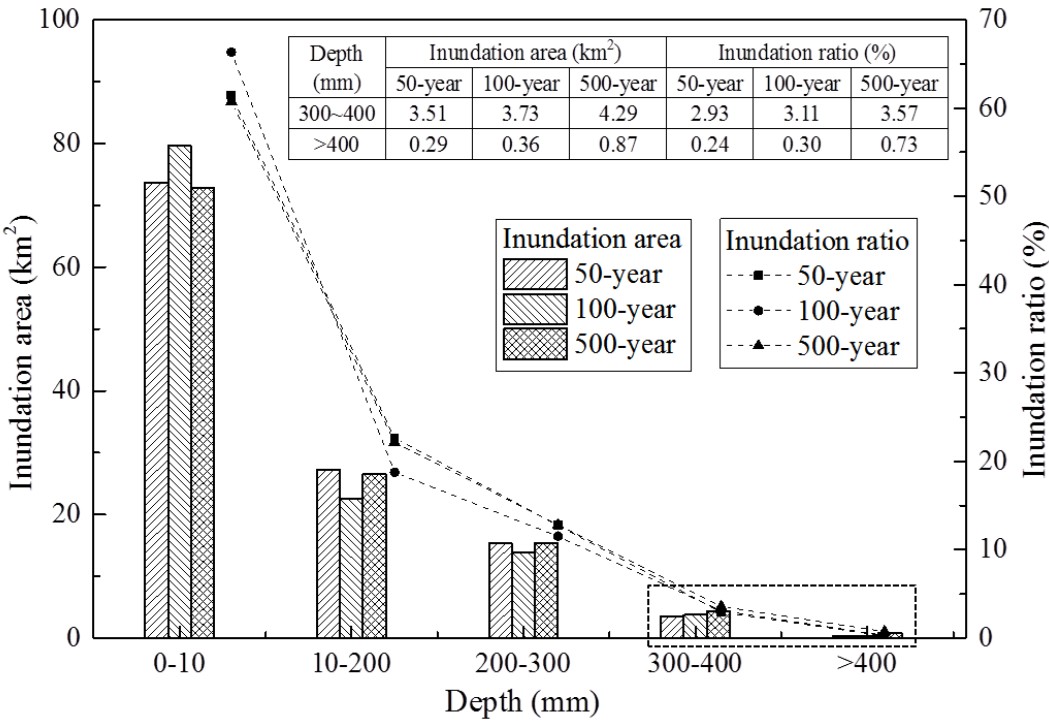

**Figure 6:** Statistical inundation area with the corresponding ratio at different depths

## 4.3 Potential inundation depth around metro station

Based on the spatial distribution of the inundation depth of a ground surface, the potential inundation depths around metro stations can be obtained with Eq. (6). Fig. 7 shows the potential inundation depths around metro stations for the 50-year rainfall intensity (Fig. 7a), 100-year rainfall intensity (Fig. 7b), and 500-year rainfall intensity scenarios (Fig. 7c). Most inundated metro stations lie in the region with a deeper flood depth. As shown in Fig. 7, the increasing rainfall intensity exacerbates the inundation depths and extents. Regarding the 50-year rainfall intensity, the Xinjiangwan Cheng Station, Yingao East Station, Yangshupu Road Station, and Longyao Road Station possibly become inundated with a depth of 100 mm. Regarding the 100-year rainfall intensity, the inundation depth of the four stations increases by 200–300 mm and the inundation expands to other central regions. In the 500-year rainfall intensity scenario, the largest inundation depth exceeds 300 mm. Other metro stations also experience inundation, with depths of 100–300 mm in the central region. In all three scenarios, the inundation initially occurs in the metro stations of Xinjiangwan Cheng, Yingao East, Yangshupu Road, and Longyao Road. Moreover, the depths increase with increasing rainfall intensity.

The number of inundated stations are presented in Fig. 7. The number of inundated metro stations increases significantly with increasing rainfall intensity. In the 500-year rainfall intensity scenario, the inundation depths of the stations of Xinjiangwan Cheng, Yingao East, Yangshupu Road, and Longyao Road are above 300 mm (see Fig. 7c).

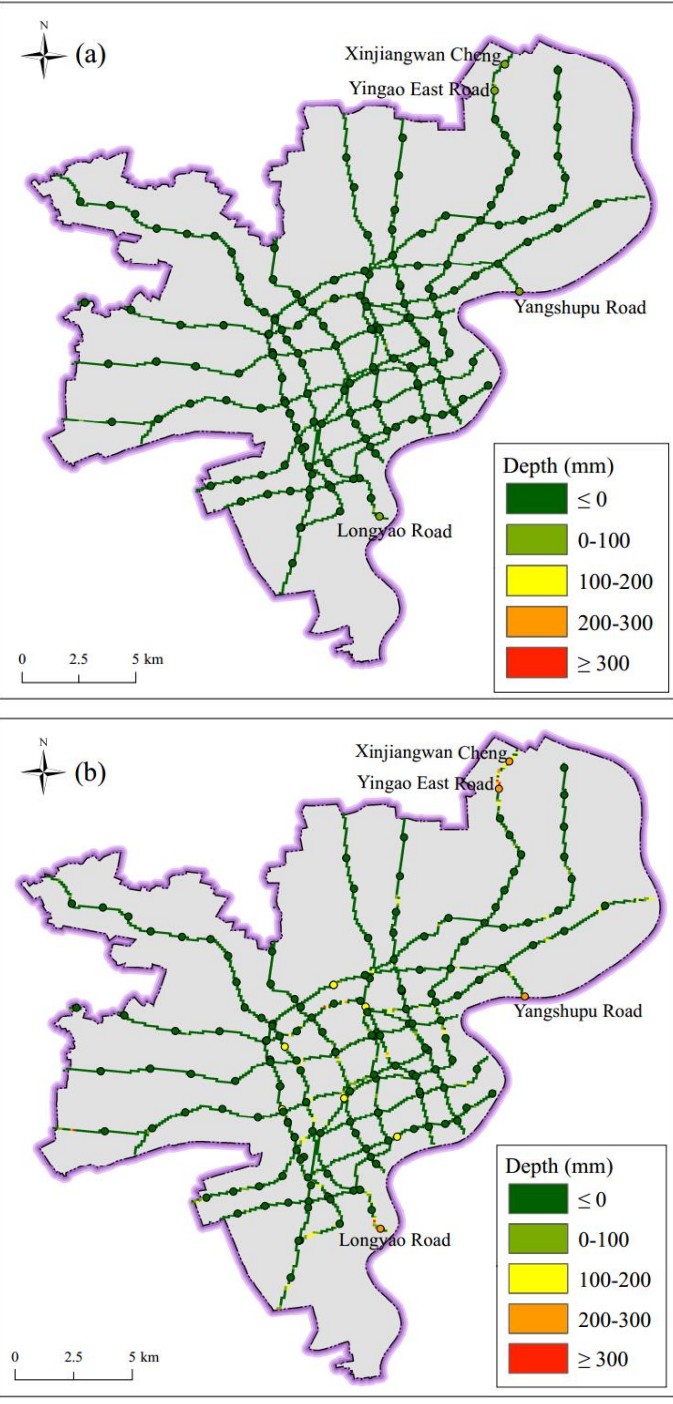

**Figure 7:** Potential inundation depth around the metro stations under different scenarios: (a) 50-year, (b) 100-year, and (c) 500-year-rainfall intensity (continuing)

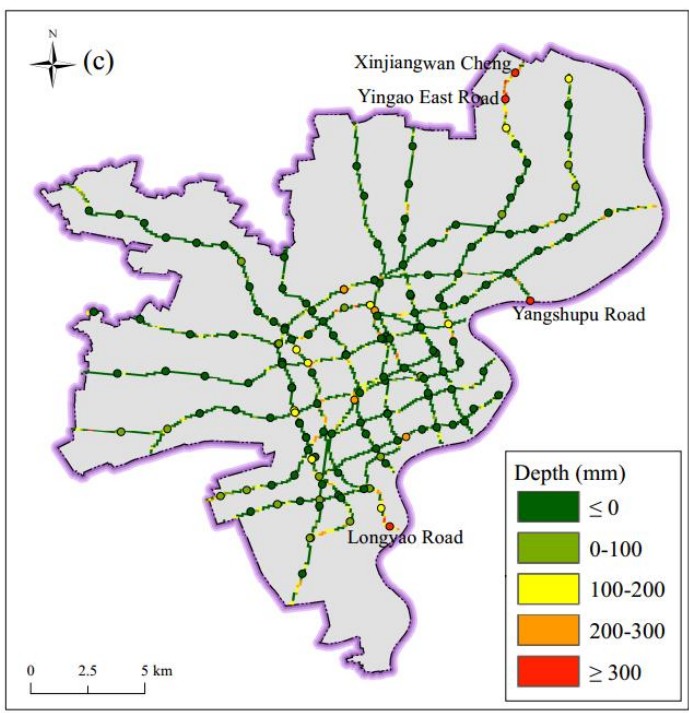

**Figure 7:** Potential inundation depth around the metro stations under different rainfall intensities: (a) 50-year, (b) 100-year, and (c) 500-year-rainfall intensity (continued)

## 4.4 Model validation

To validate the proposed model, the results of RS inundation maps (aerial or satellite) and reliable field surveys must be compared with those of the calculated inundated areas. However, the observed inundation maps of historical flood events in Shanghai are unavailable. Nevertheless, public sources can provide some historical data of inundation depths in several locations in Shanghai. Thus, the proposed model was validated by comparing the simulated data and these records. The records were provided by the following two sources: 1) flood incidents reported by public sources via websites (e.g. Google and Baidu) or in 2) publications (Huang et al., 2017; Yin et al., 2016b). The public sources provide sufficient information and include the locations of affected roads and buildings and an estimate of the inundation depth. Fig. 8 depicts the location of the recorded flood. As presented in Fig. 8a, the records cover historical floods with deep

inundation depths. Fig. 8b shows the flooding of Xujingdong Road (http://www.miss-no1.com/file/2015/08/25/618466%40152054_1.htm), and Fig. 8c presents the flooding of Yangshupu Road (http://www.chexun.com/2013-10-09/102090984_2.html). The locations of these two flood incidents correspond to the simulated flood map (see Fig. 8a).

Furthermore, publications can be used to collect recorded flood data. The official records of the rainstorm which occurred with the typhoon 'Matsa' in 2005 presented by Huang et al. (2017) are similar to those of the simulated 100-year intensity, which caused serious inundations in the districts of Yangpu, Hongkou, Changning, Putuo, and Xuhui. In addition, Yin et al. (2016b) recorded the flood of the 12 August 2011.

10    However, they investigated an area of only 3.25 km$^2$. Thus, only one validation point was extracted from the publication of Yin et al. (2016b). Finally, we collected 13 flooding locations (see Fig. 8a). Except for point 5, the points of the flood locations originate from the report of Huang et al. (2017), and point 5 originates from the paper of Yin et al. (2016b). Next, the simulated results were compared with the records of the 13 validation points. Fig. 9 shows a comparison of the inundation depths of the simulated results

15    and recorded data. For points 2–12, the simulated data agree well with the recorded data with a relative difference of less than 10%. However, the simulated depths at points 1 and 13 are much deeper than those of the records. One possible reason could be the fixed-boundary effect, because they are located near the boundary. In addition, point 5 represents the flood location of 2011, which has a higher depth than that of the simulated data. Overall, the calculated results reflect the trends of the floodwater movement and

20    depths.

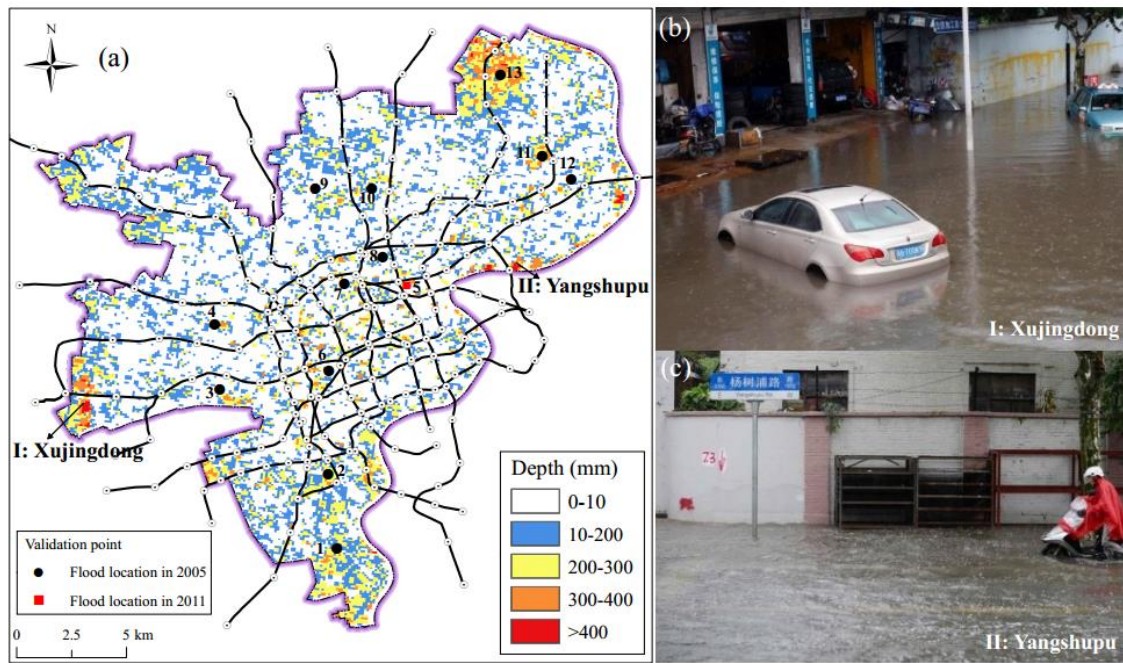

**Figure 8:** Distribution of the recorded flood locations: (a) recorded flood locations, (b) inundation of the Xujingdong road, and (c) inundation of the Yangshupu road

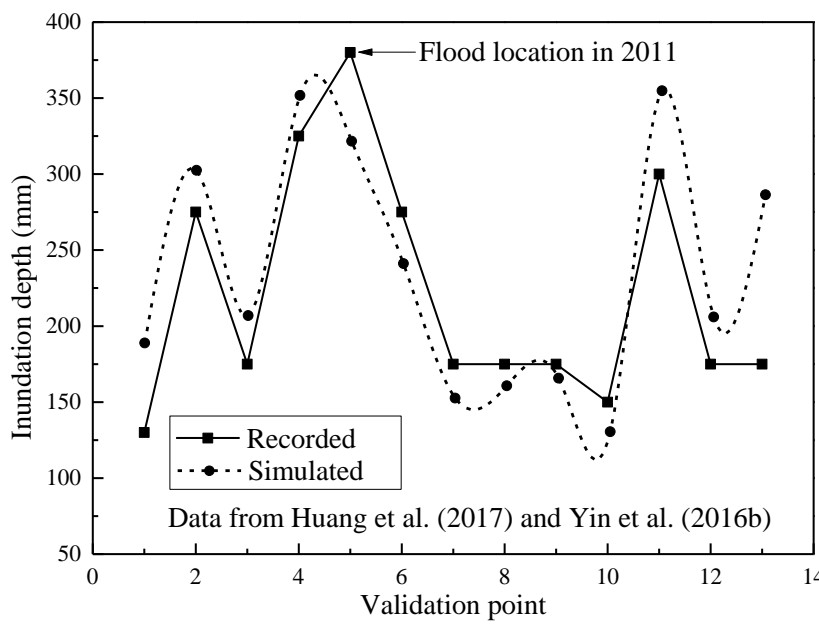

**Figure 9:** Comparison of the inundation depths obtained from the simulated results and recorded data

## 5. Discussion

### 5.1 Model evaluation and limitations

In this study, the open-source inundation model SWMM, combined with GIS, was adopted to evaluate inundation risks. To improve the approach, a new algorithm is proposed to simulate the overland flow on the ground surface. The algorithm can integrate the SWMM into the GIS. This approach can predict the inundation risks on a regional scale, whereas the existing methods can only evaluate small areas. It is assumed that the rainwater flows from one grid to another. Moreover, a spreading coefficient is used to move the runoff between neighbouring grids. During the surface flow, the rainwater is redistributed between the ground surface and drainage stations. However, the existing drainage network is not directly considered in this method owing to its complexity for a regional scale. Alternatively, the capacity of the drainage station (see Fig. 2) was used to reduce the water quantity of each subcatchment calculated in SWMM. The function of the drainage network is reflected by the drainage capacity of each drainage station. In reality, a short-term rainstorm is easy to induce flash floods in urban area. The existing researches paid more attention to flash floods induced by short-term rainstorm within 2 h or 3 h (Yin et al., 2016a, b; Wu et al., 2017). Therefore, in this study, a rainfall duration of 2 h was selected to simulate. During the simulation, the rainwater is supposed to flow on the ground surface and in the drainage stations. As already mentioned, the effects of the underground drainage network are not considered. During a short flash flood, the rainwater mainly flows on the surface (the spreading process occurs in a domain). Thus, the proposed approach is suitable for the simulation of rainstorms with short durations. Because of a lack of recorded data for the inundation depths of the metro stations, only the inundation depth on the ground surface was validated through a comparison between the simulated results and records of historical floods. The comparison reveals that the model can capture the surface flow dynamics of rainwater. However, the calculated inundation depths and validated results exhibit some differences. This can be ascribed to the uncertainties originating from various assumptions for the parametric values, data quality, and modelling

conditions. These uncertainties result in a larger inundation depth than in the recorded data. Overall, the simulated results provide a relatively reliable prediction of inundation risks. Although the simulated results reveal various uncertainties, the deviations are acceptable, and the model is suitable for urban inundation predictions.

## 5.2 Flooding prevention measures

The simulated results show a spatiotemporal distribution of the inundation profiles. The inundation profiles are characterized by a consistency in the rainfall scenarios with larger inundation depths and extents corresponding to higher rainfall intensities. In the scenario of the 500-year rainfall intensity, various regions within the study area are predicted to suffer catastrophic inundation, particularly those regions near the Huangpu River. This phenomenon may be due to the backwater effect, which is well known to be stronger and more apparent at riversides than that in inland regions. Therefore, there is a need to improve the drainage facilities (e.g., sewer system, manhole, and outlet) along the Huangpu River. Inundation of the metro system primarily occurred in the regions with a deep inundation depth. To mitigate the damage caused by inundation in metro system, the drainage capacity of the ground surface around the metro station should be increased (Suarez et al. 2005; Aoki et al. 2016). In addition, the height of the step of the metro station with a high inundation risk should be increased. Drainage facilities within the metro station should also be allocated for the emergency of flooding. In the future, more flooding adaptation measures should be taken to mitigate the catastrophic damages caused by urban flooding.

## 6. Conclusions

This paper presents a method for the evaluation of inundation risks through the integration of a hydraulic model and GIS-based analysis via a proposed algorithm. The proposed approach was used to predict the inundation risk of the metro system in Shanghai. The results were verified by recorded flooding events.

According to the results, major conclusions were drawn as follows:

(1) A new algorithm was proposed to simulate the inundation extent and depth on the ground surface. This algorithm included two aspects: i) the determination of the grid location and ii) an interactive calculation of the spreading process. With the proposed algorithm, the incorporated SWMM and GIS are adopted to yield a spatial-temporal distribution of the inundation risk on the ground surface.

(2) The study area was classified into subcatchments, and their corresponding information was stored in the GIS database. The information of each subcatchment was exported and input in the SWMM model to calculate the water volume of each subcatchment. Moreover, each subcatchment was meshed into grids. The calculated water volume was adopted to update the water level of each grid using iterative calculation with the proposed algorithm. Finally, the stable water level of each grid in GIS was used to determine the inundation depth.

(3) Based on the inundation depth on the ground surface, an equation was proposed to calculate the inundation depth of the metro system qualitatively. The proposed equation provides a quantitative evaluation of the metro system by considering the drainage capacity and characteristics of each metro station.

(4) The proposed approach was used to simulate the inundation risks of the metro stations in Shanghai under 50-year, 100-year, and 500-year intensities. The results show that the stations of Xinjiangwan Cheng, Yingao East, Yangshupu Road, and Longyao Road might become inundated. In the 50-year rainfall intensity, these four stations will be inundated with a depth of 100 mm. In the 100-year rainfall intensity, the inundation depth of the four stations increases by 200–300 mm, whereas the inundation expands to other central regions. In the 500-year rainfall intensity, the highest inundation depth exceeds 300 mm. Moreover, other metro stations experience inundation with depths of 100–300 mm in the central region.

*Acknowledgements*. The research work described herein was funded by the National Natural Science

Foundation of China (NSFC) (Grant No. 41672259) and the Innovative Research Funding of the Science and Technology Commission of Shanghai Municipality (Grant No. 18DZ1201102). These financial supports are gratefully acknowledged.

5    *Author contribution.* This paper represents a result of collaborative teamwork. Shui-Long Shen developed the concept; Hai-Min Lyu drafted the manuscript; Jun Yang provided constructive suggestions and revised the manuscript; Zhen-Yu Yin collected the data and revised the manuscript. The four authors contributed equally to this work.

10   *Competing interests.* The authors declare that they have no conflict of interest.

**Appendix: Pseudo-code of the algorithm for the spreading procedure**

---

**Algorithm**: Algorithm for the spreading process of the runoff volume.

**input:** Arcgis.in $\in$ (*A, E, h, x, y*)

      ! *Data with area, elevation, average water depth, and X/Y coordinates from the arcgis database.*

**output:** Data.out $\in$ (*A, h`*)

      ! *Water depth of each grid.*

Determine the relative location and spreading coefficient of each grid around the target grid.

Spreading process

**Do** i = 1, N

  ! *N is the iteration step of the spreading steps.*

  Rank of spreading for each grid

$$Q_{t\arg et} = \sum_{i=1}^{link} h_i A_i$$

  ! *Based on the water quantity of each grid, select the target grid.*

  Start spreading

**Do** n = 1, M

  ! *M is the total number of spreading grids. The maximum value of M=8.*

  **If** ( (**abs**($h_I$-$h_{Ij}$)>0.01) .and. $Q_I$>0) **Then**

$$Q_{diffuse} = \sum_{j=1}^{n} (h_I - h_{Ij}) \cdot A_j \quad (j=1,2,\cdots,n)$$

  ! *Based on the spreading coefficient, allocate the water quantity and update the water level of each grid around the target.*

**End if**

**End do**

**End do**

---

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

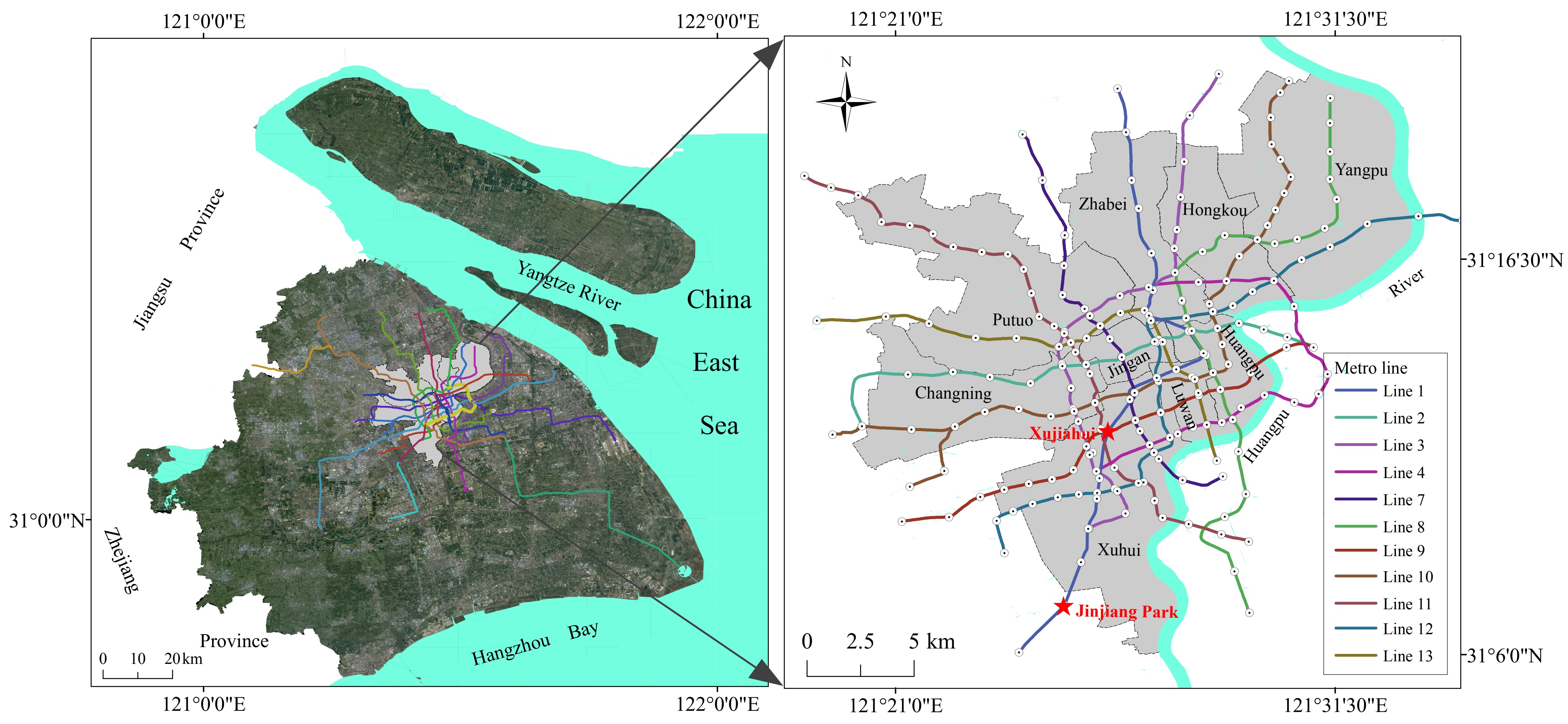

Fig. 1 Metro line distribution in study area of Shanghai

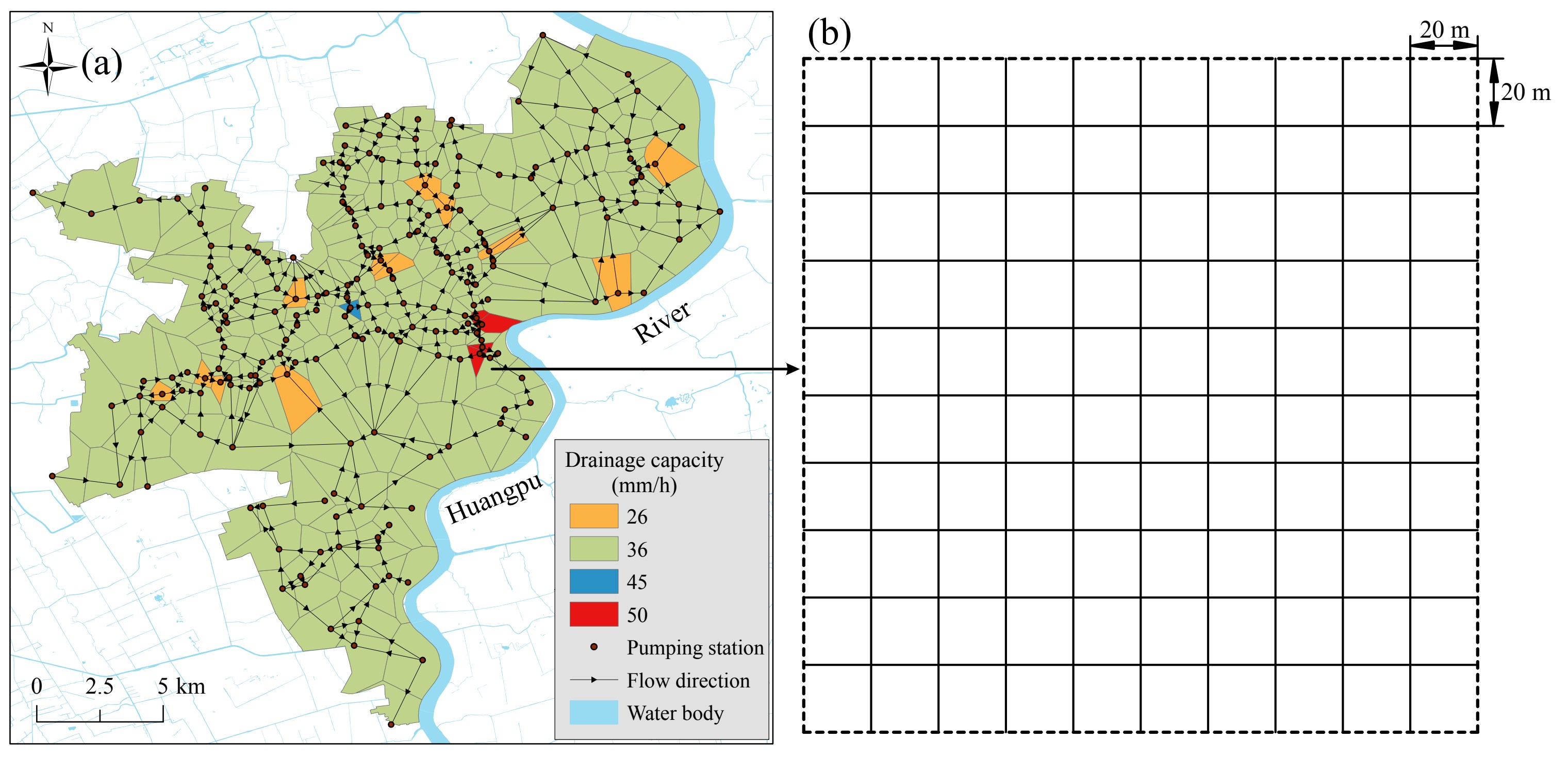

Fig. 2 Subcatchment and calculated grid: (a) driange capacity and flow direction
of each subcatchment; (b) calibrated grid of each sucatchment

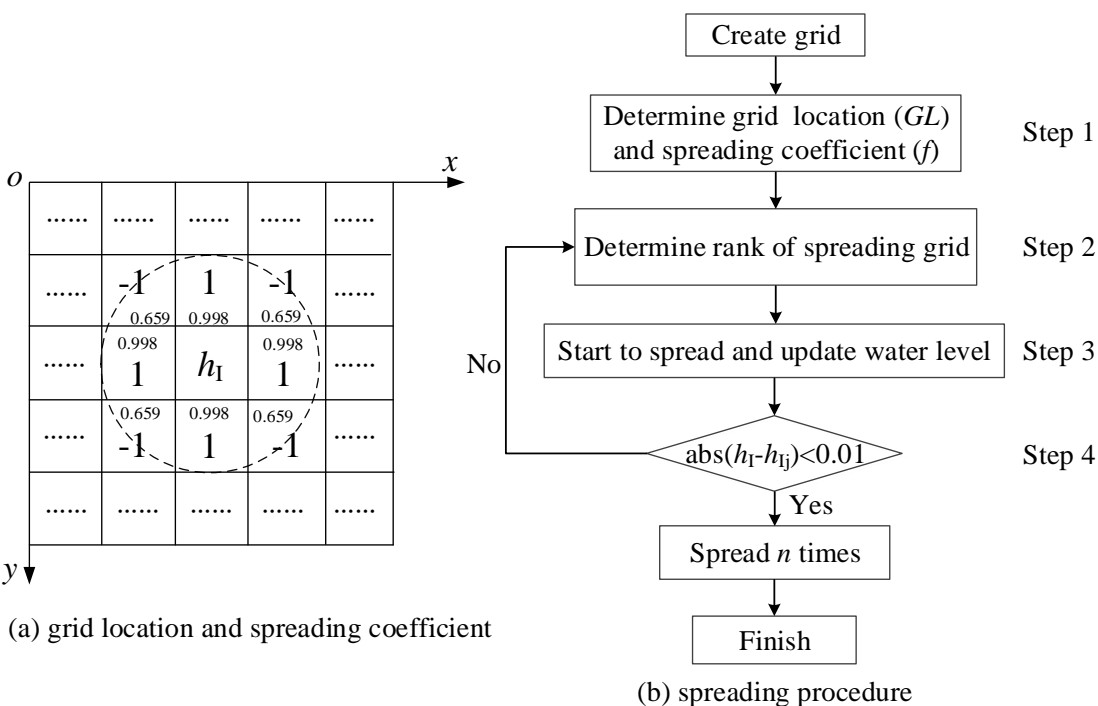

(a) grid location and spreading coefficient

(b) spreading procedure

**Figure 3:** Description of the spreading process: (a) determination of the grid location and spreading coefficient and (b) iterative calculation of the spreading process

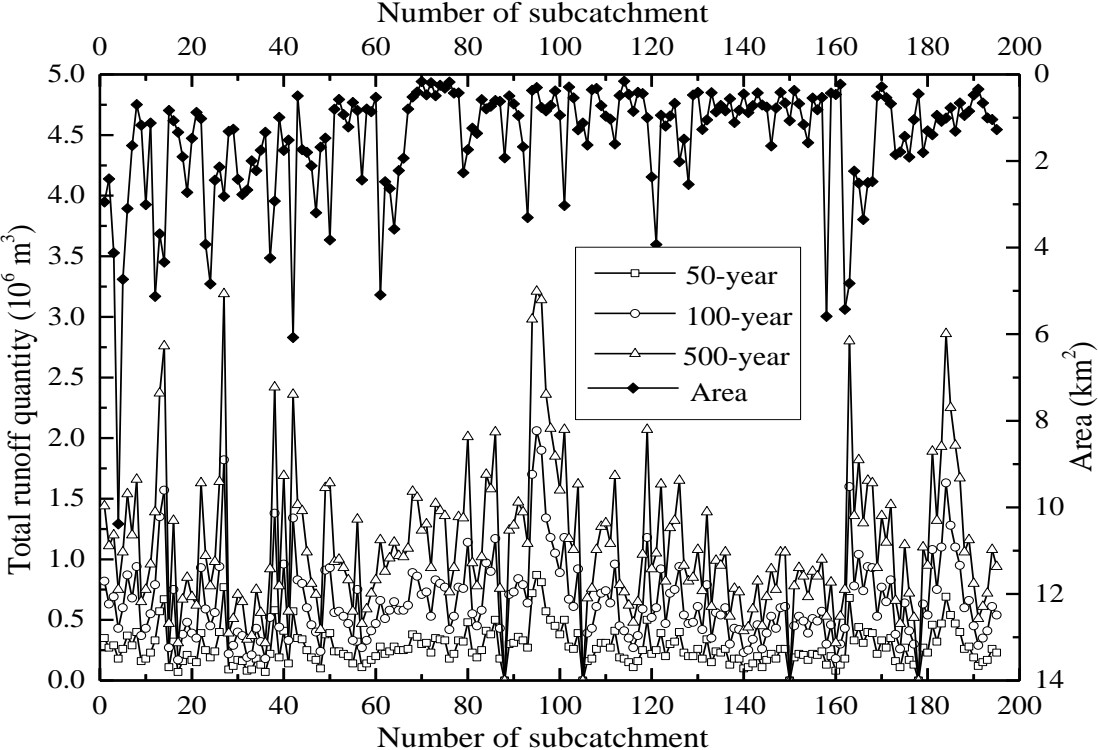

**Figure 4:** Runoff volume of each subcatchment in the corresponding area under different scenarios

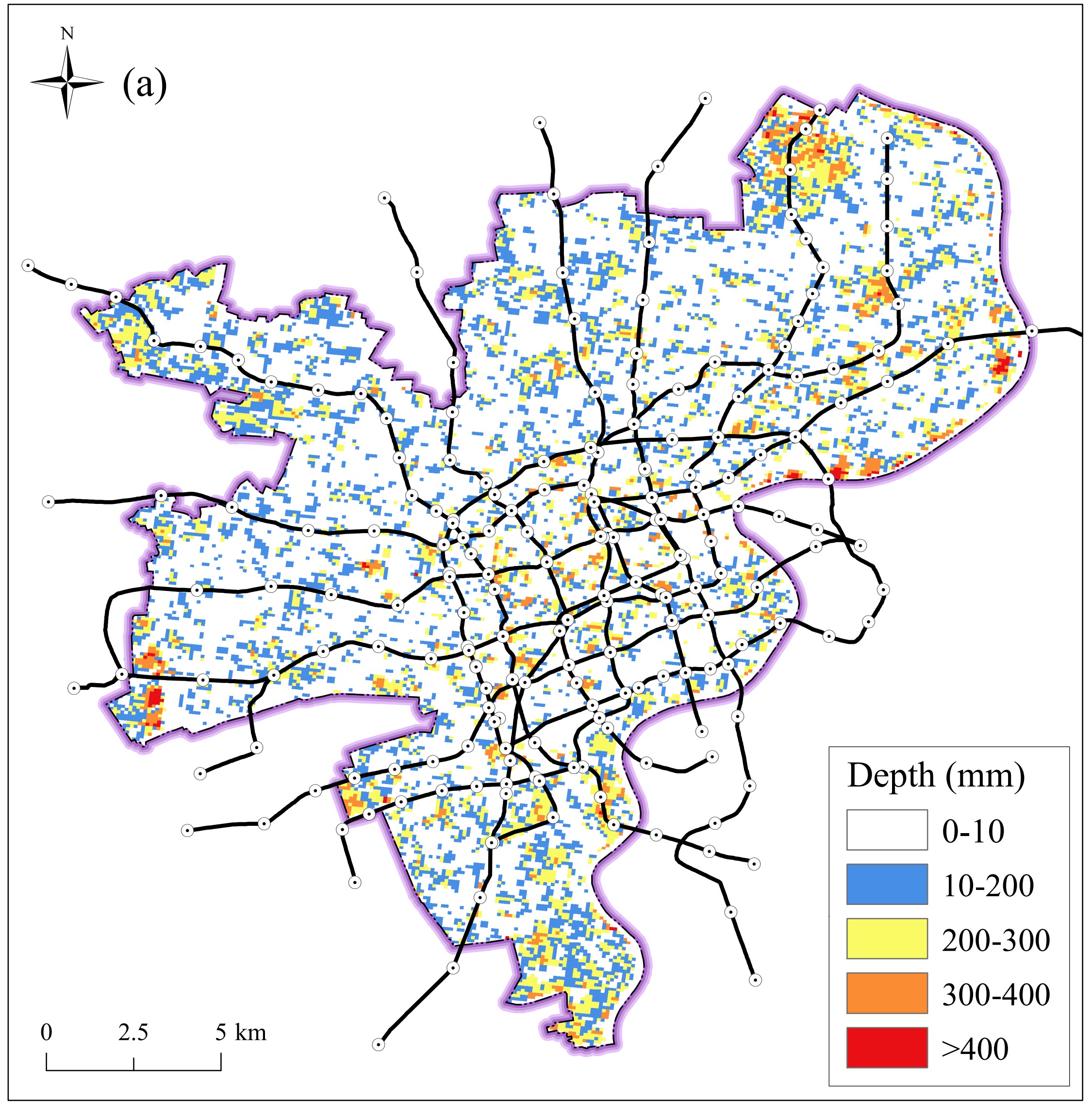

Fig. 5 Distribution of the potential inundation extent and depth under different rainfall intensity: (a) 50-year, (b) 100-year, and (c) 500-year-rainfall intensity

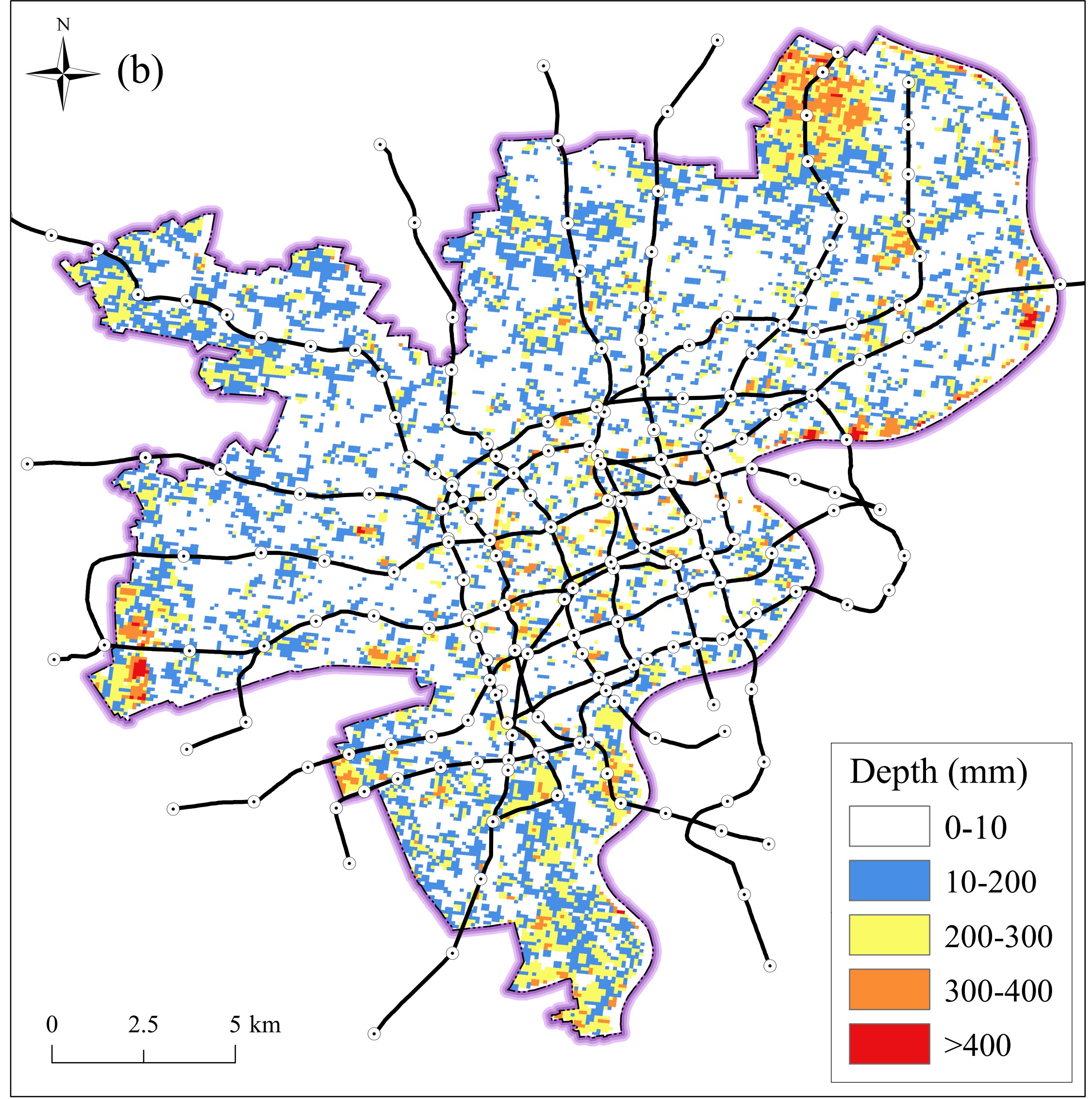

Fig. 5  Distribution of the potential inundation extent and depth under different rainfall intensity: (a) 50-year, (b) 100-year, and (c) 500-year-rainfall intensity

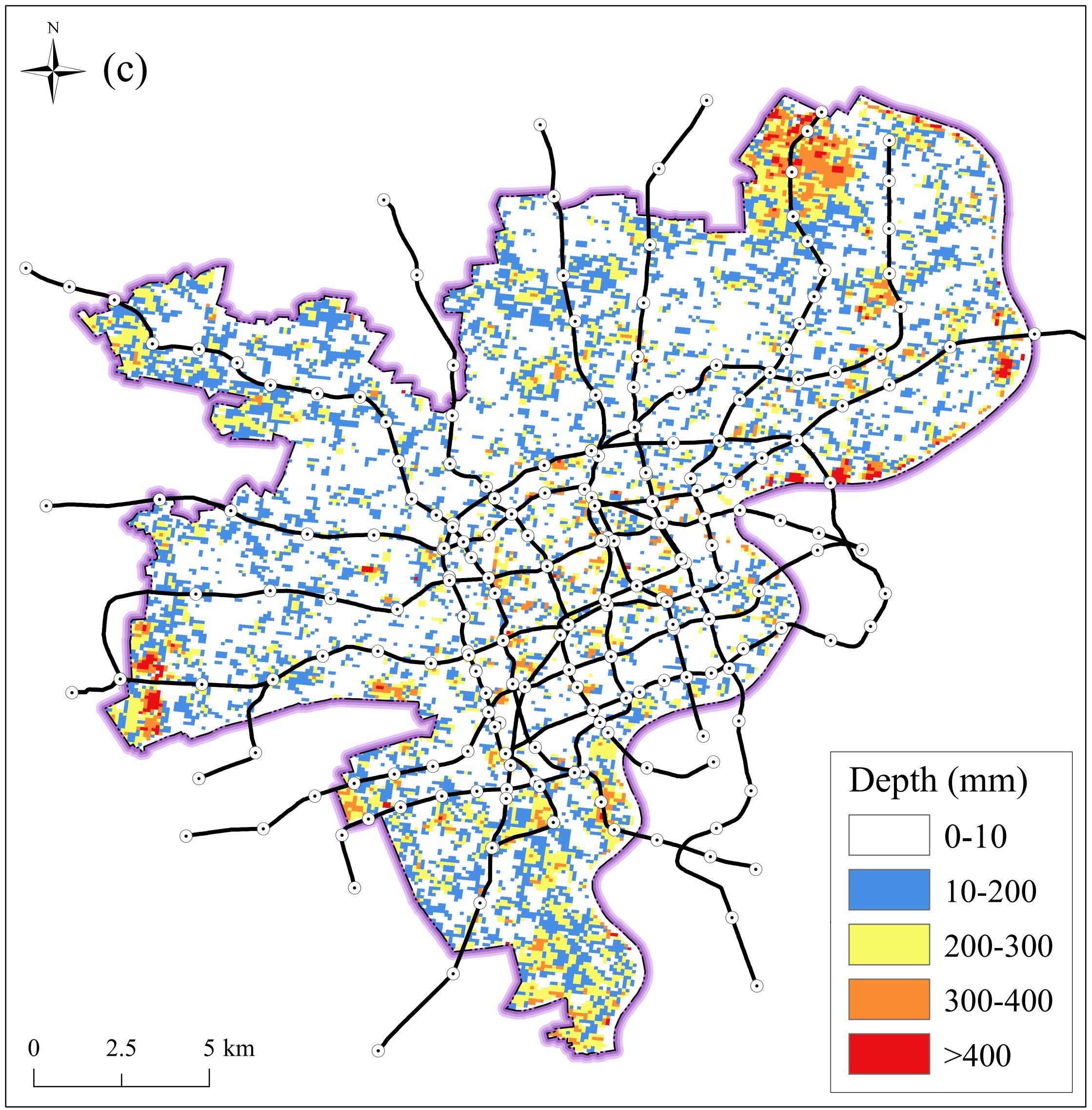

Fig. 5  Distribution of the potential inundation extent and depth under different rainfall intensity: (a) 50-year, (b) 100-year, and (c) 500-year-rainfall intensity

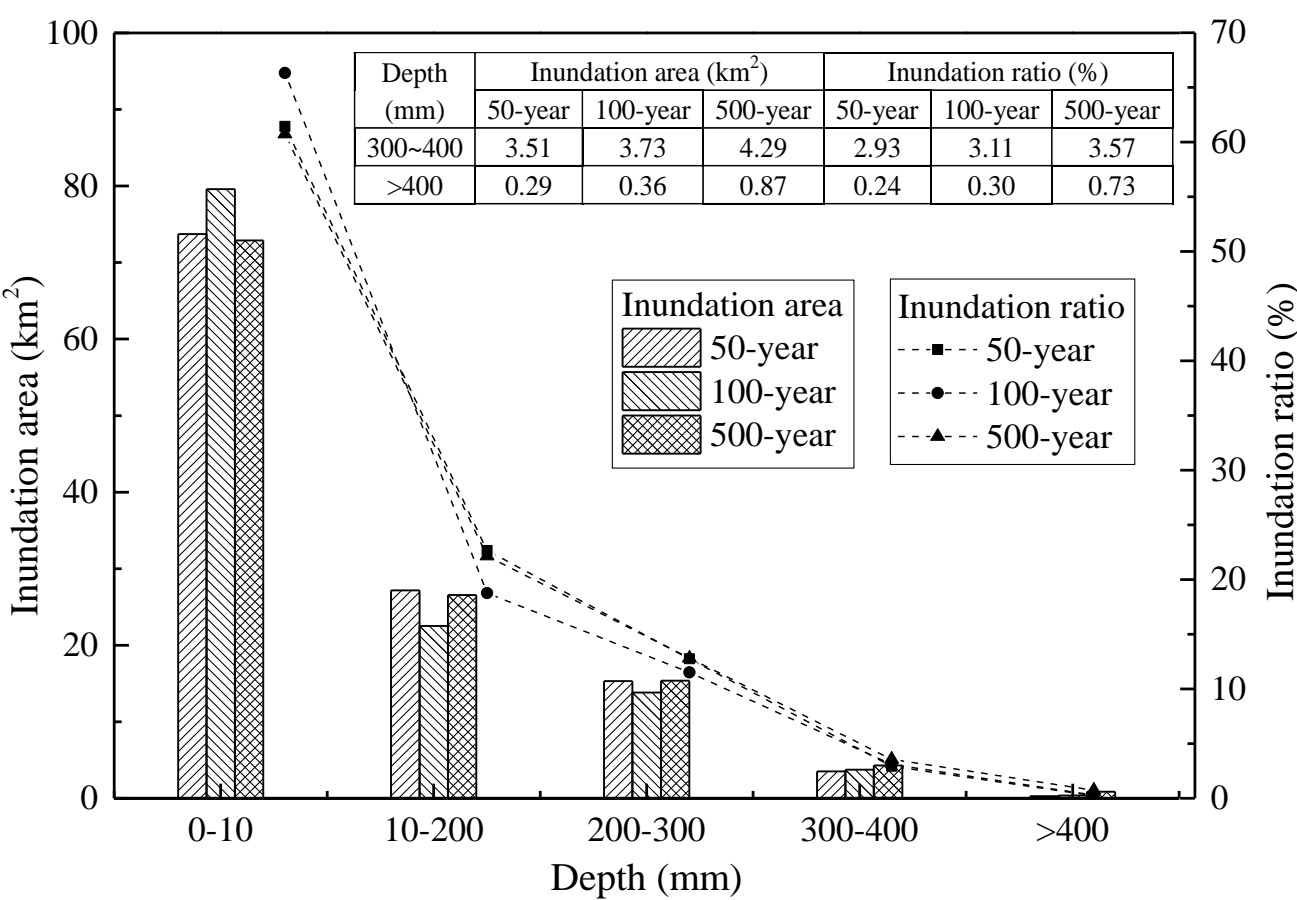

| Depth | Inundation area (km$^2$) | | | Inundation ratio (%) | | |
|---|---|---|---|---|---|---|
| (mm) | 50-year | 100-year | 500-year | 50-year | 100-year | 500-year |
| 300~400 | 3.51 | 3.73 | 4.29 | 2.93 | 3.11 | 3.57 |
| >400 | 0.29 | 0.36 | 0.87 | 0.24 | 0.30 | 0.73 |

Fig. 6 Statistical inundation area with the corresponding ratio at different depths

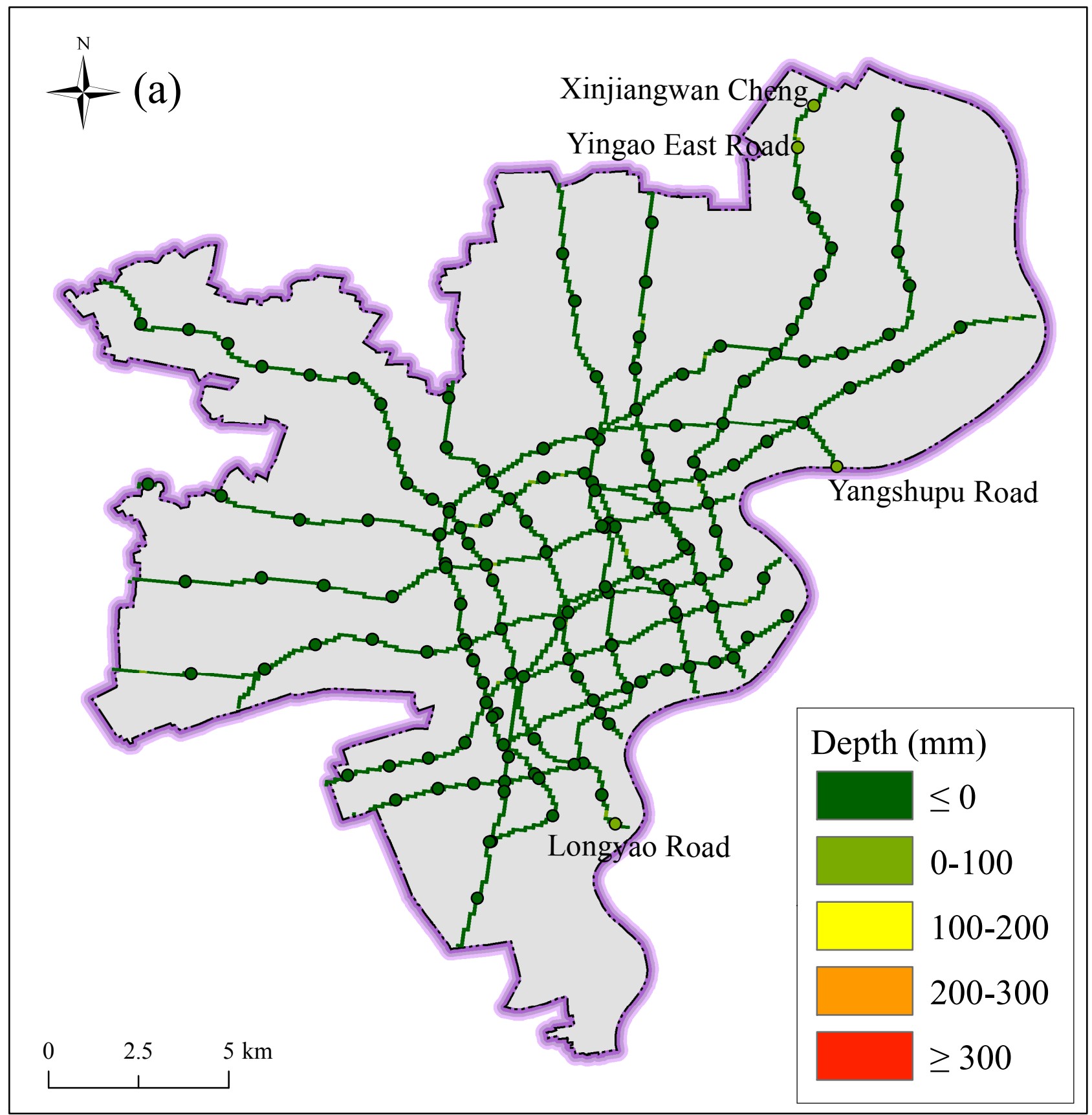

Figure 7: Potential inundation depth around the metro stations under different rainfall intensities: (a) 50-year, (b) 100-year, and (c) 500-year-rainfall intensity

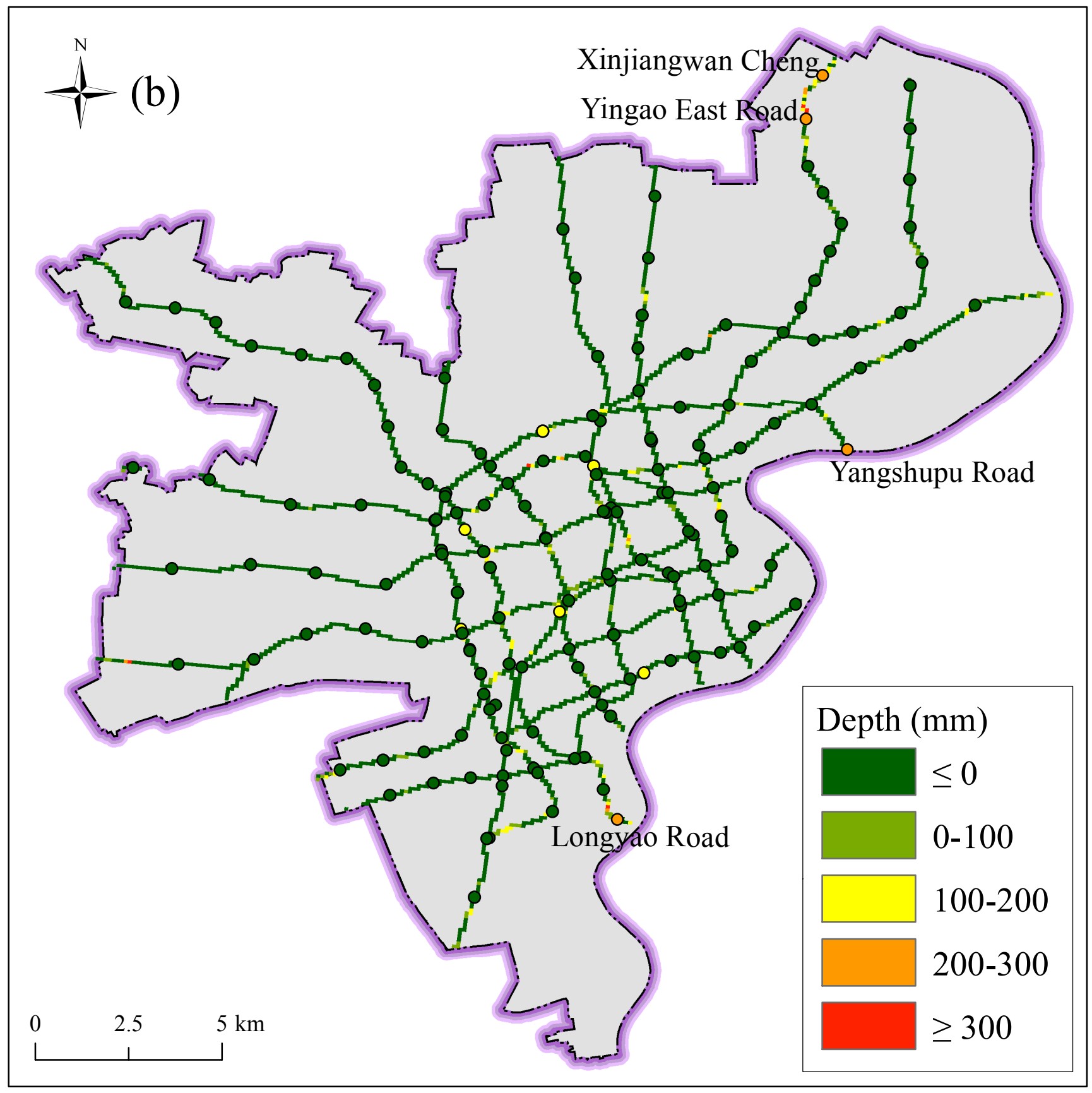

Figure 7: Potential inundation depth around the metro stations under different rainfall intensities: (a) 50-year, (b) 100-year, and (c) 500-year-rainfall intensity

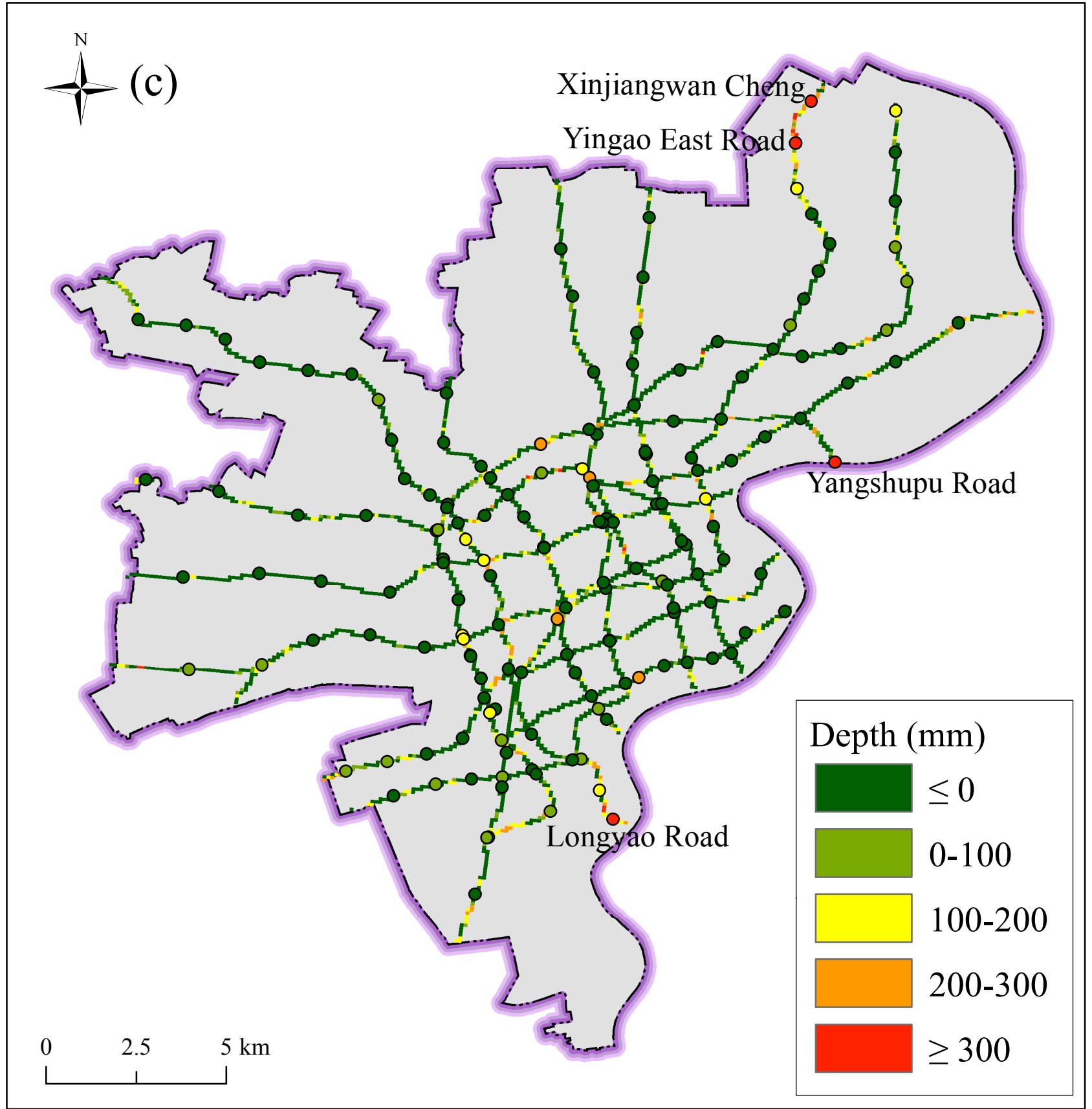

Figure 7: Potential inundation depth around the metro stations under different rainfall intensities: (a) 50-year, (b) 100-year, and (c) 500-year-rainfall intensity

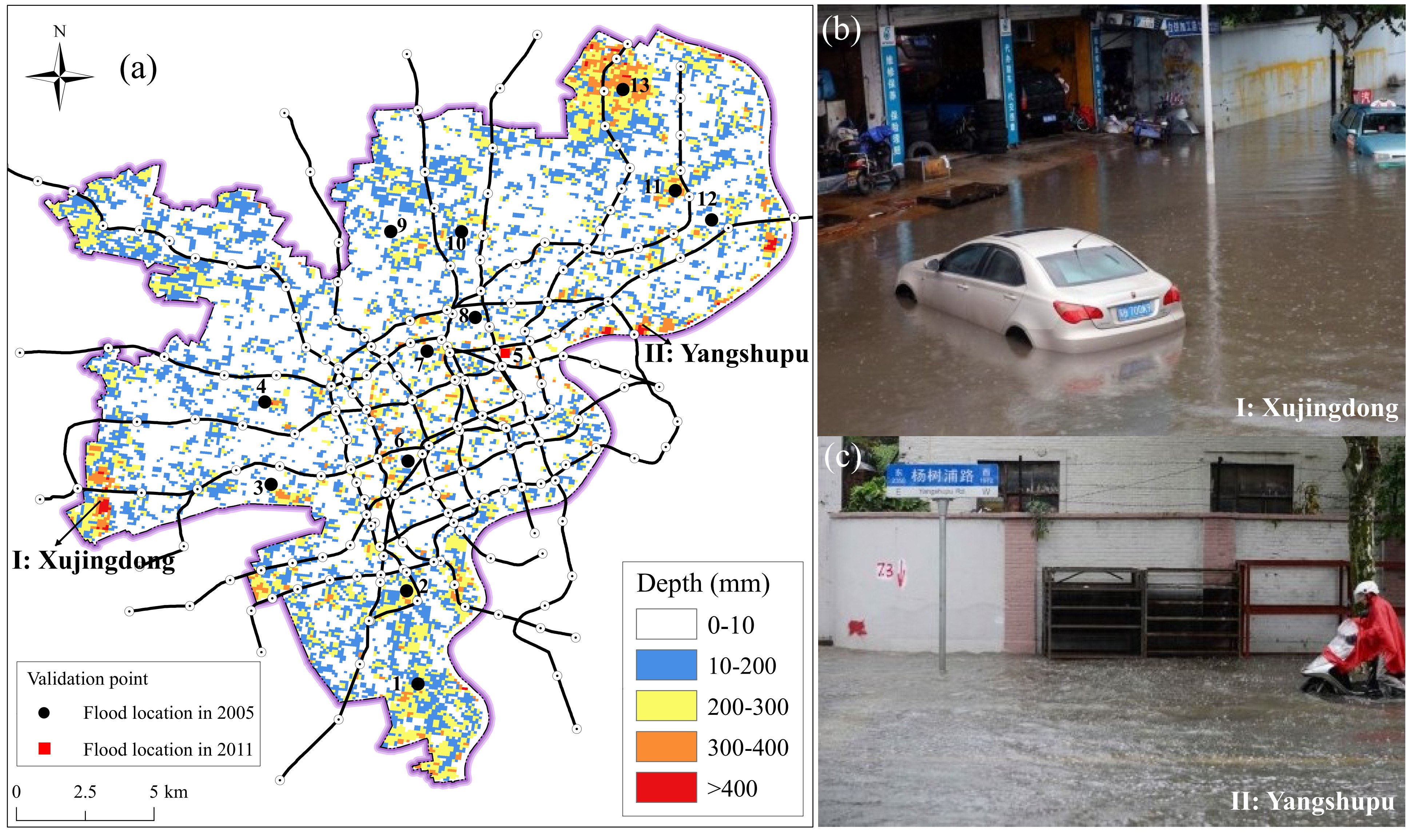

Fig. 8  Distribution of the recorded flood locations: a) recorded flood locations, b) inundation of the
Xujingdong road, and c) inundation of the Yangshupu road

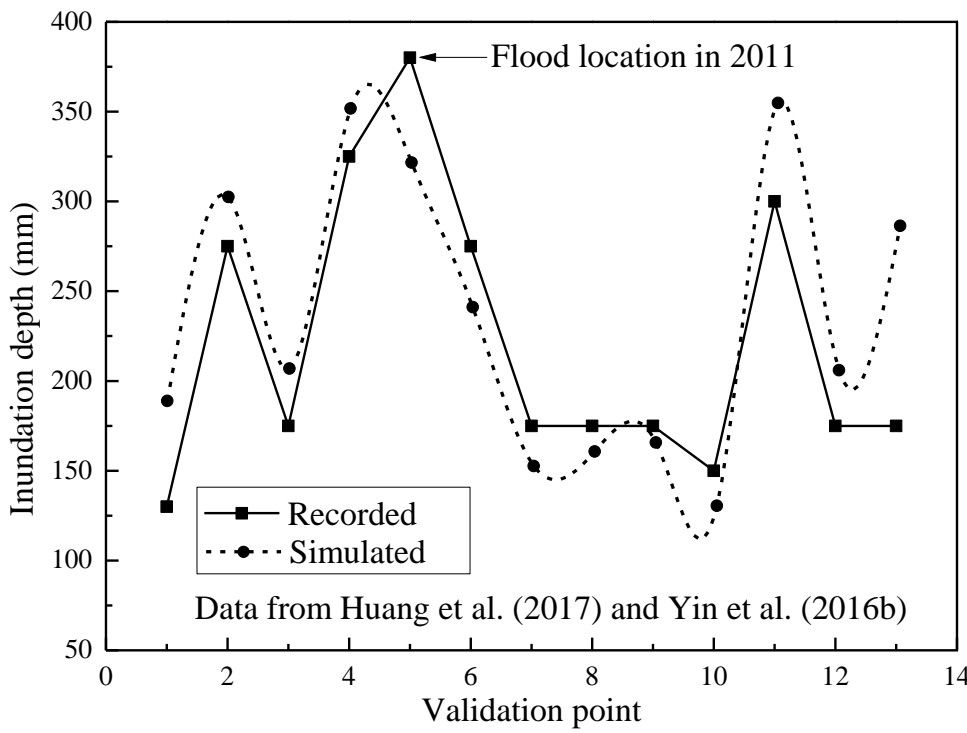

**Figure 9:** Comparison of the inundation depths obtained from the simulated results and recorded data

**Table 1** Parameters of the subcatchments in the SWMM

| Parameter | Meaning | Value |
|---|---|---|
| Area (km$^2$) | Area of each subcatchment | 10.38–0.16 |
| Width (m) | Width of each subcatchment | 5283.83–432.45 |
| Impervious (%) | Percentage of the impervious area | 65–80 |
| Slope (°) | Average slope of each subcatchment | 0.3–5.5 |
| Destore-impervious (mm) | Depression storage depth in the impervious area | 1.5 |
| Destore-pervious (mm) | Depression storage depth in the previous area | 5 |
| N-impervious | Manning's coefficient in the impervious area | 0.1 |
| N-pervious | Manning's coefficient in the previous area | 0.24 |
| MaxRate (mm/h) | Maximum infiltration rate | 72 |
| MinRate (mm/h) | Minimum infiltration rate | 0.72 |
| Decay (h$^{-1}$) | Decay constant | 4 |
| Dry (d) | Drying time | 2 |

**Appendix: Pseudo-code of the algorithm for the spreading procedure**

---

**Algorithm**: Algorithm for the spreading process of the runoff volume.

**input:** Arcgis.in € ($A, E, h, x, y$)
! *Data with area, elevation, average water depth, and X/Y coordinates from the arcgis database.*
**output:** Data.out € ($A, h$`)
! *Water depth of each grid.*

Determine the relative location and spreading coefficient of each grid around the target grid.

Spreading process
**Do** i = 1, N
! *N is the iteration step of the spreading steps.*

Rank of spreading for each grid

$$Q_{t \arg et} = \sum_{i=1}^{link} h_i A_i$$

! *Based on the water quantity of each grid, select the target grid.*

Start spreading
**Do** n = 1, M
! *M is the total number of spreading grids. The maximum value of M=8.*
**If** ( (**abs**($h_I$-$h_{Ij}$)>0.01) .and. $Q_I$>0) **Then**

$$Q_{diffuse} = \sum_{j=1}^{n} (h_I - h_{Ij}) \cdot A_j \quad (j=1,2,\cdots,n)$$

! *Based on the spreading coefficient, allocate the water quantity and update the water level of each grid around the target.*
**End if**
**End do**
**End do**