# Peer review of "Inundation analysis of metro systems with SWMM incorporated in GIS: a case study in Shanghai"

_Hydrology and Earth System Sciences, 2019_

## Short Comment (SC1) · 4 Apr 2019

Review Comments Title of Paper: Scenario-based inundation analysis of metro systems: a case study in Shanghai

Manuscript Number: hess-2019-28

This article proposed an approach to investigate the flood risk to metro system. This is an interesting topic for urban management to improve flood resilience of significant infrastructure. To achieve the objective, the metro system of Shanghai is studied. The authors have used SWMM software to simulate the scenario and developed an algorithm to calculate the inundation depth. The results are very useful for the metro management and decision-making for municipal government. However, the following

weak points have to be addressed.

GENERAL COMMENTS: - The equations have to be clearly referenced. - A brief summary of the simulated scenario should be added to the introduction. - It seems that some repeat information is provided in section to make it lengthy otherwise. SECTION 2.2 - value of r is taken as 0.45. how is this value determined? - Type of soil in the study area should also be added so as to get a clear image of the study site. This is an important point missed. - Do assumptions made in methodology are validated for a similar type of work at another site also? SECTION 3.1.2 - Description showing the that calculation of width and area should be included. SECTION 3.2 - Step 4 in the flowchart of figure 3(b), needs to be clarified. - In Result and Analysis: If using abbreviation anywhere, its full form needs to be stated at first. - In discussion: Bit more flooding prevention measures and tips should be added in this section. Does the type of soil have any effect on inundation depth? - Details about 50 years rainfall intensity and 100 years rainfall intensity should also be included in conclusion as the whole manuscript covers 50, 100 and 500 years rainfall intensity. The following publications may be useful for this article: - An enhanced inundation method for urban flood hazard mapping at the large catchment scale. Journal of Hydrology, 2019, 571: 873-882. - The effectiveness of low-impact development for urban inundation risk mitigation under different scenarios: a case study in Shenzhen, China. Natural Hazards and Earth System Science, 18, 2525–2536, 2018, https://doi.org/10.5194/nhess-18-2525-2018 - Modelling urban floods and drainage using SWMM and MIKE URBAN: a case study. Natural Hazards, 2016, 84(2): 749-776. - Urbanization and climate change impacts on future urban flooding in Can Tho city, Vietnam. Hydrology and Earth System Science, 17, 379-394, 2013, https://doi.org/10.5194/hess-17-379-2013

---

## Short Comment (SC2) · 16 Apr 2019

This manuscript presented an investigation on the flooding risk of metro system, especially for the stations' inundation during severe rainstorm. The topic is interesting and within the scope of this journal. Overall quality of this manuscript is well. However, to improve the quality and readability of the manuscript, suggestions should be considered by the authors.

Following comments would help the authors;

(1) Page 1 in line 18; suggest to rephrase the sentence structure "In addition, an equation is proposed to qualitatively calculate the inundation to figure out possible inundation risks of Shanghai metro system;

[Figure]

(2) Page 4; in Introduction: "Begin with a broad, general statement of the topic and narrow it down to the context". Please add some introductory lines of Shanghai metro system regarding inundation or overflow problems in the past, before the objectives of the research;

(3) Page 8 in line 4; suggest to specify the tools of GIS used to measure inundation depth;

(4) Page 8 in line 17; please revise "was" with "is";

(5) Page 9 in line 8; which tool in GIS is used to extract sub catchments, please mention it;

(6) Page 14: in Fig. 4, note that SI unit for runoff is m3/s, please recheck the unit you used is ok?

(7) Page 18 in line 18; which four metro stations are found under high inundations risks. please specify it in your manuscript.

(8) Page 22; Fig. 8, please note flood location in 2005 in the map is indicated with circular simple but in legend it is triangular, revise the legends

(9) Page 24; please make sure that all 3-objectives discussed in "introduction" section, are achieved in the "conclusion" as well. It would be good to add inundation depth and results for all 3 scenarios as mentioned in objectives of the study

---

## Referee Comment (RC1) · Malek Alshorman (Referee) · 23 Apr 2019

-page3, line 20-25: SWMM can be used for large catchments please refer to the following paper:https://link.springer.com/article/10.1007%2Fs41207-018-0092-7. -page10, table-1: the maximum and minimum infiltration rates which used in the model is default values in SWMM, we have to use the exact values for the catchment area and based on geotechnical information. -page 17, figure-6: are you consider the existing drainage network in the modeling? if yes you have to show that in the text it no you have to consider it because it will have a significant effect.

---

## Short Comment (SC3) · 12 May 2019

This article proposed an approach to investigate the flood risk to metro system. This is an interesting topic for urban management to improve flood resilience of significant infrastructure. To achieve the objective, the metro system of Shanghai is studied. The authors have used SWMM software to simulate the scenario and developed an algorithm to calculate the inundation depth. The results are very useful for the metro management and decision-making for municipal government. However, the following weak points have to be addressed.

Answer: Thanks for the reviewer's positive and suggestive comments. We have revised the manuscript according to the comments point by point.

[Figure]

GENERAL COMMENTS: - The equations have to be clearly referenced.

Answer: Thanks for the reviewer's comment. The Eqs. (1) to (3) have been referenced. Other equations are proposed by authors.

- A brief summary of the simulated scenario should be added to the introduction. It seems that some repeat information is provided in section to make it lengthy otherwise.

Answer: We have summarized the scenario-based inundation analysis from line 1 to line 6 in page 3.

Line 1-6 in page 3: Scenario-based inundation analysis presents inundation risk under different scenarios (Willems 2013; Naulin et al. 2013), which requires the topography, land-use, and urban drainage system data. Owing to the complex interaction between the drainage system and overland surface in urban regions, scenario-based models can only simulate inundation over a small range, e.g., less than 3 km2 (Wu et al. 2017), which limits their application. Thus, the application of scenario-based model needs to be extended to the problem of overland flow over a large scale, e.g. whole region with area over several hundred square kilometers.

SECTION 2.2 - value of r is taken as 0.45. how is this value determined?

Answer: Thanks for the reviewer's comment. The value of r is an empirical value. We refereed the related publication to determine is as 0.45. We have added the reference in context in line 11 in page 7.

Reference: Yin, J., Yu, D.P., Yin, Z.E., Liu, M., and He, Q.: Evaluating the impact and risk of pluvial flash flood on intra-urban road network: A case study in the city center of Shanghai, China. Journal of Hydrology. 537, 138-145, doi: 10.1016/j.jhydrol.2016.03.037, 2016a.

- Type of soil in the study area should also be added so as to get a clear image of the study site. This is an important point missed. Answer: Thanks for the reviewer's comment. The soil type of the study area has been added in the revised manuscript.

The study area is located in urban center, where the dense buildings exist. The blocking effects have important influence on surface flow. We have added this section from line 6 to line 25 in page 10.

Line 6-25 in page 10: The impervious parameter was determined based on the types of land use. The study area is located in urban centre, where the land use has no big changes. The existence of dense buildings in the study area makes more than 80% of the surface is impervious. Due to the existence of road pavement, subgrade and many municipal pipelines under the road, the water infiltration through road and subsurface under road is very small, which can be considered as impervious. Thus, soil infiltration and evapotranspiration have slight effects on surface runoff concentration during short-term flash flooding under rainstorm. The soil infiltration mainly depends on green land (combined by lawn, flower bed, and grove) and water body within the study area. In this aspect, the geotechnical information in Shanghai is as follows. The groundwater table is higher than 2 m below ground surface. The soil type at the depth 2 m is a mixed soil with of sand (5%), silt (55%), and clay (40%) according to Shanghai Geotechnical Investigation Code (DGJ08-37-2012). At the surface, sand content increased to 15%, so that soil has the hydraulic conductivity of 2ïĆť10-5 m/s, which is 72 mm/h; at the bottom of water body, the soil has more clay content (>50%) and less sand content (<5%) with the hydraulic conductivity of 2ïĆť10-7 m/s, which is 0.72 mm/h (Shen et al., 2015). According to the SWMM handbook, the maximum infiltration rate is determined as 72 mm/h to reflect the characteristics of green land, while the minimum value is 0.72 mm/h to reflect the characteristics of water body, since the soil under water body is saturated clay. In addition, the blocking effects of the buildings have significant influences on the surface runoff generation and concentration. Therefore, the heights of the existing buildings were extracted to modify the elevation of the calculated grids, which have crucial influence on the redistribution of rainwater during calculation.

- Do assumptions made in methodology are validated for a similar type of work at another site also?

Answer: The proposed algorithm of surface flow spreading is based on the variation of grid elevation. Thus, the assumptions in the algorithm is suitable to simulate the spreading of rainwater in flat region. When the study area with large difference between high and low elevations, the rainwater will be converged in low region. We have added discussion from line 1 to line 3 in page 24.

Line 1-3 in page 24: The proposed algorithm is used to spreading surface flow based on the variation of the elevation in study area. Thus, the proposed approach is suitable to simulate the inundation risk in flat region.

SECTION 3.1.2 - Description showing the that calculation of width and area should be included.

Answer: Thanks for the reviewer's comment. The width and area of each subcatchment are obtained using GIS tools.

SECTION 3.2 - Step 4 in the flowchart of figure 3(b), needs to be clarified.

Answer: Thanks for the reviewer's comment. We have revised this section from line 7 to line 10 in page 11.

Line 7-10 in page 11: Fig. 3 shows the description of the spreading procedure of runoff. Fig. 3(a) illustrates the determination of grid location and spreading coefficient. Fig 3(b) is the iterative calculation of the spreading process. Firstly, grids are created with 20 m×20 m meshes across the study area using GIS fishnet tools [see Fig. 3(a)]; secondly, the calculated average inundation depth is extracted from each grid [see Fig. 3(b)].

- In Result and Analysis: If using abbreviation anywhere, its full form needs to be stated at first.

Answer: Thanks for the reviewer's detailed comment. We have added full form of abbreviation in the context.

- In discussion: Bit more flooding prevention measures and tips should be added in this section. Does the type of soil have any effect on inundation depth?

Answer: Thanks for the reviewer's comment. We have added the flooding prevention measures in the section of 'Flood prevention measures'. The effects of the soil type on inundation depth have been added from line 6 to line 25 in page 10.

- Details about 50 years rainfall intensity and 100 years rainfall intensity should also be included in conclusion as the whole manuscript covers 50, 100 and 500 years rainfall intensity.

Answer: Thanks for the reviewer's comment. We have added the results of 50-year and 100-year rainfall intensity in conclusion from line 18 to line 24 in page 25.

Line 18-24 in page 25: (3) The proposed approach was used to simulate the inundation risk of the metro stations in Shanghai under 50-year, 100-year, and 500-year-scenarios. The results showed that these stations of Xinjiangwan Cheng, Yingao east, Yangshupu Road, and Longyao Road are possible to inundated. In the 50-year-rainfall intensity, these four stations are predicted to be inundated at 100 mm-depth. In the 100-year-rainfall intensity, the inundation depth of the four stations increased by 200–300 mm, whereas the inundation extent exacerbated to other central regions. In the 500-year-rainfall intensity, the largest inundation depth exceeds 300 mm, and other metro stations also undergo inundation with a depth of 100–300 mm in the central region.

The following publications may be useful for this article: - An enhanced inundation method for urban flood hazard mapping at the large catchment scale. Journal of Hydrology, 2019, 571: 873-882. - The effectiveness of low-impact development for urban inundation risk mitigation under different scenarios: a case study in Shenzhen, China. Natural Hazards and Earth System Science, 18, 2525–2536, 2018, https://doi.org/10.5194/nhess-18-2525-2018 - Modelling urban floods and drainage using SWMM and MIKE URBAN: a case study. Natural Hazards, 2016, 84(2):

[Figure]

749-776. - Urbanization and climate change impacts on future urban flooding in Can Tho city, Vietnam. Hydrology and Earth System Science, 17, 379-394, 2013, https://doi.org/10.5194/hess-17-379-2013

Answer: Thanks for the reviewer's suggestive comment. We have referred the following reference both in context and reference list. Theses references are helpful to this manuscript.

Reference: Zhao, G., Xu, Z.X., Pang, B., Tu, T.B., Xu, L.Y., Du, L.G.: An enhanced inundation method for urban flood hazard mapping at the large catchment scale. Journal of Hydrology. 571: 873-882, 2019. Wu, J.S., Yang, R., Song, J.: Effectiveness of low-impact development for urban inundation risk mitigation under different scenarios: a case study in Shenzhen, China. Natural Hazards and Earth System Science. 18: 2525-2018, 2018. Bisht, D.S, Chatterjee, C., Kalakoti, S., Upadhyay, P., Sahoo, M., Panda, A.: Modeling urban floods and drainage using SWMM and MIKE URBAN: a case study. Natural Hazards. 84: 749-776, 2016. Huong, H.T.L., Pathirana, A.: Urbanization and climate change impacts on future urban flooding in Can Tho city, Vietnam. Hydrology and Earth System Science, 17: 379-394, 2013.

Please also note the supplement to this comment:
https://www.hydrol-earth-syst-sci-discuss.net/hess-2019-28/hess-2019-28-SC3-supplement.pdf

---

## Author Comment (AC1) · 12 May 2019

This manuscript presented an investigation on the flooding risk of metro system, especially for the stations' inundation during severe rainstorm. The topic is interesting and within the scope of this journal. Overall quality of this manuscript is well. However, to improve the quality and readability of the manuscript, suggestions should be considered by the authors. Following comments would help the authors; (1) Page 1 in line 18; suggest to rephrase the sentence structure "In addition, an equation is proposed to qualitatively calculate the inundation to figure out possible inundation risks of Shanghai metro system;

Answer: Thanks for the reviewer's suggestive comment. We have revised this sentence

from line 17 to line 18 in page 1.

Line 17-18 in page 1: In addition, an equation is proposed to qualitatively calculate the inundation around a metro station to predict the potential inundation risks of metro system.

(2) Page 4; in Introduction: "Begin with a broad, general statement of the topic and narrow it down to the context". Please add some introductory lines of Shanghai metro system regarding inundation or overflow problems in the past, before the objectives of the research;

Answer: Thanks for the reviewer's suggestive comment. We have revised this section from line 10 to line 11 in page 2.

Line 10-11 in page 2: Numerous metro lines were inundated during the flood season (May to September) in 2016 in China, such as the metro lines in Guangzhou and Wuhan. The Shanghai Station of metro line No.1 was inundated on October 3, 2016 (Lyu et al. 2018a, b).

Reference: Lyu, H.M., Sun, W.J., Shen, S.L., and Arulrajah, A.: Flood risk assessment in metro systems of mega-cities using a GIS-based modeling approach. Science of the Total Environment, 626, 1012-1025. doi: 10.1016/j.scitotenv.2018.01.138, 2018a. Lyu, H.M., Xu, Y.S., Cheng, W.C., and Arulrajah, A.: Flooding hazards across southern China and prospective sustainability measures. Sustainability, 10(5), 1682. doi:10.3390/su10051682, 2018b.

(3) Page 8 in line 4; suggest to specify the tools of GIS used to measure inundation depth;

Answer: Thanks for the reviewer's comment. GIS is an analysis tool in this study, we use the tools (e.g., grid calculation, extraction analysis and extract multi values to points, etc.) to analyze the inundation risk. We have revised the sentence from line 1 to line 2 in page 8.

Line 1-2 in page 8: The third phase, the inundation depth around a metro station was obtained using the proposed equation and GIS tools (e.g., grid calculation, and extract multi values to points, etc)

(4) Page 8 in line 17; please revise "was" with "is";

Answer: Thanks for the reviewer's comment. It has been revised as "is" in line 18 in page 8.

(5) Page 9 in line 8; which tool in GIS is used to extract sub catchments, please mention it;

Answer: Thanks for the reviewer's comment. The elevation can be extracted using extract analysis in GIS. It has been specified in line 5 in page 8.

(6) Page 14: in Fig. 4, note that SI unit for runoff is m3/s, please recheck the unit you used is ok?

Answer: Thanks for the reviewer's comment. Here we use the total water quality of each subcatchment. Thus, the unit is m3.

(7) Page 18 in line 18; which four metro stations are found under high inundations risks. please specify it in your manuscript.

Answer: Thanks for the reviewer's comment. We have revised this sentence from line 8 to line 9 in page 19. Line 8-9 in page 19: The number of inundated stations can also be accounted from Fig. 7. It is clearly seen that with the increase in the rainfall intensity, the number of inundated metro stations is increasing. For the 500-year-rainfall intensity, the inundation depth of these stations of Xinjiangwan Cheng, Yingao east, Yangshupu Road, and Longyao Road over 300 mm (see Fig. 7c).

(8) Page 22; Fig. 8, please note flood location in 2005 in the map is indicated with circular simple but in legend it is triangular, revise the legends

Answer: Thanks for the reviewer's detailed comment. We have revised Fig. 8.

(9) Page 24; please make sure that all 3-objectives discussed in "introduction" section, are achieved in the "conclusion" as well. It would be good to add inundation depth and results for all 3 scenarios as mentioned in objectives of the study

Answer: Thanks for the reviewer's comment. We have added the inundation depth and results for all 3 scenarios in conclusions from line 18 to line 24 in page 25.

Line 18-24 in page 25: (3) The proposed approach was used to simulate the inundation risk of the metro stations in Shanghai under 50-year, 100-year, and 500-year-scenarios. The results showed that these stations of Xinjiangwan Cheng, Yingao east, Yangshupu Road, and Longyao Road are possible to inundated. In the 50-year-rainfall intensity, these four stations are predicted to be inundated at 100 mm-depth. In the 100-year-rainfall intensity, the inundation depth of the four stations increased by 200–300 mm, whereas the inundation extent exacerbated to other central regions. In the 500-year-rainfall intensity, the largest inundation depth exceeds 300 mm, and other metro stations also undergo inundation with a depth of 100–300 mm in the central region.

Please also note the supplement to this comment:
https://www.hydrol-earth-syst-sci-discuss.net/hess-2019-28/hess-2019-28-AC1-supplement.pdf

———————————————

[Figure]

Fig. 8  Distribution of the recorded flood locations: a) recorded flood locations, b) inundation of the
Xujingdong road, and c) inundation of the Yangshupu road

**Fig. 1.** Figure 8: Distribution of the recorded flood locations: a) recorded flood locations, b)
inundation of the Xujingdong road, and c) inundation of the Yangshupu road

---

## Author Comment (AC2) · 12 May 2019

-page3, line 20-25: SWMM can be used for large catchments please refer to the following paper: https://link.springer.com/article/10.1007%2Fs41207-018-0092-7. Answer: Thanks for the suggestive comment. We have referred the following reference in the revised manuscript. This is a helpful publication.

Reference: Ai-Mashaqbeh, O., Shorman, M. (2019). Modeling of the stormwater runoff quantity and quality in Amman Zarqua Basin, Jordan. Euro-Mediterranean Journal for Environmental Integration, https://doi.org/10.1007/s41207-018-0092-7.

-page10, table-1: the maximum and minimum infiltration rates which used in the model is default values in SWMM, we have to use the exact values for the catchment area

and based on geotechnical information. Answer: Thanks for the reviewer's constructive comment. This aspect was revised according to the comment. The study area is located in urban center, where the land use has no big changes. The dense buildings exist in the study area, where more than 80% of the surface is impervious. Due to the existence of road pavement, subgrade and many municipal pipelines under the road, the water infiltration through road and subsurface under road is very small, which can be considered as impervious. Thus, soil infiltration and evapotranspiration have slight effects on surface runoff concentration during short-term flash flooding under rainstorm. The soil infiltration mainly depends on green land (combined by lawn, flower bed, and grove) and water body within the study area. In this aspect, the geotechnical information in Shanghai is as follows. The groundwater table is higher than 2 m below ground surface. The soil type at the depth 2 m is a mixed soil with of sand (5%), silt (55%), and clay (40%) according to Shanghai Geotechnical Investigation Code (DGJ08-37-2012). At the surface, sand content increased to 15%, so that the soil has the hydraulic conductivity of 2ïĆť10-5 m/s, which is 72 mm/h; at the bottom of water body, the soil has more clay (>50%) and less sand (<5%) with the hydraulic conductivity of 2ïĆť10-7 m/s, which is 0.72 mm/h (Shen et al., 2015). According to the SWMM handbook, the maximum infiltration rate is determined as 72 mm/h to reflect the characteristics of green land, while the minimum value is 0.72 mm/h to reflect the characteristics of water body. Moreover, in the study area, the blocking effects of the existing buildings have significant effects on the surface runoff generation and concentration. Therefore, the height of the existing building has been paid more attention during surface runoff redistribution. We have rewritten the section of the parameters in SWMM from line 6 to line 25 in page 10.

Reference: DGJ08-37-2012. (2012). Code for investigation of geotechnical engineering in Shanghai. Shanghai Urban Construction and Communications Commission, Shanghai. (in Chinese) Shen, S.L., Wang, J.P., Wu, H.N., Xu, Y.S., Ye, G.L., and Yin, Z.Y. (2015). Evaluation of hydraulic conductivity for both marine and deltaic deposits based on piezocone testing. Ocean Engineering, 110(2015), 174-182. doi:

10.1016/j.oceaneng.2015.10.011

Line 6-25 in page 10: The impervious parameter was determined based on the types of land use. The study area is located in urban centre, where the land use has no big changes. The existence of dense buildings in the study area makes more than 80% of the surface is impervious. Due to the existence of road pavement, subgrade and many municipal pipelines under the road, the water infiltration through road and subsurface under road is very small, which can be considered as impervious. Thus, soil infiltration and evapotranspiration have slight effects on surface runoff concentration during short-term flash flooding under rainstorm. The soil infiltration mainly depends on green land (combined by lawn, flower bed, and grove) and water body within the study area. In this aspect, the geotechnical information in Shanghai is as follows. The groundwater table is higher than 2 m below ground surface. The soil type at the depth 2 m is a mixed soil with of sand (5%), silt (55%), and clay (40%) according to Shanghai Geotechnical Investigation Code (DGJ08-37-2012). At the surface, sand content increased to 15%, so that the soil has the hydraulic conductivity of $2 \times 10^{-5}$ m/s, which is 72 mm/h; at the bottom of water body, the soil has more clay content (>50%) and less sand content (<5%) with the hydraulic conductivity of $2 \times 10^{-7}$ m/s, which is 0.72 mm/h (Shen et al., 2015). According to the SWMM handbook, the maximum infiltration rate is determined as 72 mm/h to reflect the characteristics of green land, while the minimum value is 0.72 mm/h to reflect the characteristics of water body, since the soil under water body is saturated clay. In addition, the blocking effects of the buildings have significant influences on the surface runoff generation and concentration. Therefore, the heights of the existing buildings were extracted to modify the elevation of the calculated grids, which have crucial influence on the redistribution of rainwater during calculation.

-page 17, figure-6: are you consider the existing drainage network in the modeling? if yes you have to show that in the text it no you have to consider it because it will have a significant effect. Answer: Thanks for the reviewer's constructive comment. We agree that the drainage network has significant effects on simulated results. Indeed,

we haven't modelled the existing drainage network directly. It is difficult to consider the drainage network directly in the model since the drainage network data is difficult to collect in such a large study area with 120 km2. Thus, if model each drainage, the calculation in such large area become time consuming. To solve this problem, we use the capacity of drainage station to reflect the function of drainage network. Fig. 2 shows the capacity of drainage station. In this model, we suppose that the rainwater is flowed from one subcatchment to another. During surface flowing, the rainwater is redistributed between ground surface and drainage station. The water quantity calculated by SWMM model is reduced by corresponding drainage capacity of each subcatchment (see Fig. 2). The reduced water quantity of each subcatchment was used to redistribute in the algorithm modelling. Moreover, we use a spreading coefficient to express the spreading process. The spreading coefficient is used for moving runoff between neighbor subcatchments. We have revised this section from line 14 to line 16 in page 8, and we have added discussions of limitations for the proposed approach from line 10 to line 15 in page 23, and line 1 to line 3 in page 24.

Line 14-16 in page 8: Moreover, the function of drainage network is reflected by the drainage capacity of each drainage station (see Fig. 2). The water quantity of each subcatchment calculated in SWMM is reduced by the capacity of the drainage station.

Line 10-15 in page 23 and Line 1-3 in page 24: It is supposed that the rainwater is flowed from one subcatchment to another. Moreover, a spreading coefficient is used for moving runoff between neighbouring subcatchments. During surface flowing, the rainwater is redistributed between ground surface and drainage station. The limitation of the integrated approach is that the existing drainage network has not been directly considered during simulation, since the complexity of the drainage network in a regional scale. Alternatively, the capacity of the drainage station (see Fig. 2) is used to reduce the water quantity of each subcatchment calculated in SWMM. The function of drainage network is reflected by the drainage capacity of each drainage station.

Figure 2: Calculated subcatchment and grid in SWMM and GIS: (a) drainage capacity

and flow direction of each subcatchment; (b) calculated grid of each subcatchment

Please also note the supplement to this comment:
https://www.hydrol-earth-syst-sci-discuss.net/hess-2019-28/hess-2019-28-AC2-supplement.pdf

---

## Author Comment (AC4) · 12 May 2019

[revised manuscript text omitted]

**3.1.2 Model input and determination of parameters**

Based on the aforementioned method of subcatchment division, each subcatchment was assigned with its own topographical characteristics. The model included 195 subcatchments and 204 junctions. Each subcatchment in the SWMM model included the parameters of width, area, and permeability. The width and area can be calculated by GIS tools. Table 1 tabulates the parameters of the subcatchments in the SWMM. The impervious parameter was determined based on the types of land use. The study area is located in urban centre, where the land use has no big changes. The existence of dense buildings in the study area makes more than 80% of the surface is impervious. Due to the existence of road pavement, subgrade and many municipal pipelines under the road, the water infiltration through road and subsurface under road is very small, which can be considered as impervious. Thus, soil infiltration and evapotranspiration have slight effects on surface runoff concentration during short-term flash flooding under rainstorm. The soil infiltration mainly depends on green land (combined by lawn, flower bed, and grove) and water body within the study area. In this aspect, the geotechnical information in Shanghai is as follows. The groundwater table is higher than 2 m below ground surface. The soil type at the depth 2 m is a mixed soil with of sand (5%), silt (55%), and clay (40%) according to Shanghai Geotechnical Investigation Code (DGJ08-37-2012). At the surface, sand content increased to 15%, so that soil has the hydraulic conductivity of $2\times10^{-5}$ m/s, which is 72 mm/h; at the bottom of water body, the soil has more clay content (>50%) and less sand content (<5%) with the hydraulic conductivity of $2\times10^{-7}$ m/s, which is 0.72 mm/h (Shen et al., 2015). According to the SWMM handbook, the maximum infiltration rate is determined as 72 mm/h to reflect the characteristics of green land, while the minimum value is 0.72 mm/h to reflect the characteristics of water body, since the soil under water body is saturated clay. In addition, the blocking effects of the buildings have significant influences on the surface runoff generation and concentration. Therefore, the heights of the existing buildings were extracted to modify the elevation of the calculated grids, which have crucial influence on the redistribution of rainwater during calculation.

**Table 1** Parameters of the subcatchments in the SWMM

| Parameter | Meaning | Value |
|---|---|---|
| Area (km²) | Area of each subcatchment | 10.38–0.16 |
| Width (m) | Width of each subcatchment | 5283.83–432.45 |
| Impervious (%) | Percentage of the impervious area | 65–80 |
| Slope (°) | Average slope of each subcatchment | 0.3–5.5 |
| Destore-impervious (mm) | Depression storage depth in the impervious area | 1.5 |
| Destore-pervious (mm) | Depression storage depth in the previous area | 5 |
| N-impervious | Manning's coefficient in the impervious area | 0.1 |
| N-pervious | Manning's coefficient in the previous area | 0.24 |
| MaxRate (mm/h) | Maximum infiltration rate | 72 |
| MinRate (mm/h) | Minimum infiltration rate | 0.72 |
| Decay (h⁻¹) | Decay constant | 4 |
| Dry (d) | Drying time | 2 |

**3.2 Data conversion between GIS and SWMM**

Following the calibration of runoff volume of each subcatchment, the next step is to determine the spreading procedure of the calibrated runoff. The spreading procedure algorithm is used to integrate the data between GIS and SWMM. Fig. 3 shows the description of the spreading procedure of runoff. Fig. 3(a) illustrates the determination of grid location and spreading coefficient. Fig 3(b) is the iterative calculation of the spreading process. First, grids are created with 20 m×20 
[revised manuscript text omitted]
 neighbouring subcatchments. During surface flowing, the rainwater is redistributed between ground surface and drainage station. The limitation of the integrated approach is that the existing drainage network has not directly considered during simulation, since the complexity of the drainage network in a regional scale. Alternatively, the capacity of the drainage station (see Fig. 2) is used to reduce the water quantity of each subcatchment calculated in SWMM. The function

of drainage network is reflected by the drainage capacity of each drainage station. The proposed algorithm is used to spreading surface flow based on the variation of the elevation in study area. Thus, the proposed approach is suitable to simulate the inundation risk in flat region. 
[revised manuscript text omitted]

---

## Author Comment (AC6) · 12 May 2019

The supplement is the revised manuscript.

Please also note the supplement to this comment:
https://www.hydrol-earth-syst-sci-discuss.net/hess-2019-28/hess-2019-28-AC6-supplement.pdf
* * *

---

## Referee Comment (RC2) · Anonymous Referee #2 · 6 Jun 2019

The topic of the manuscript is interesting and relevant. However, I have strong concerns about the proposed methodology because (perhaps) its poor description in the manuscript: - By including "Scenario-based..." in the title of the manuscript I was expecting something else than considering rainfall events of different return periods - this is classic in hydrology and I would not consider it a "scenario-based" analysis.

- The way the EPA SWMM model is connected to the GIS "model" is not clear. Also, EPA SWMM has two main parts: hydrology (lumped catchments) and hydraulics (pipes). How can EPA SWMM be used to estimate water depth on the terrain surface (see e.g. Page 4, line 8)? from EPA SWMM simulations one can obtain "flooding" results in each model node (representing e.g. a manhole), but it is a flow rate and not a depth (for the reasons indicated above).

[Figure]

- The literature cited in the manuscript is rather old. For example, the authors cite the 2002 study from Horrit and Bates. More than 15 years have passed since this study was published and significant developments in terms of computational power have occurred. The authors should include more recent studies that might contradict their argument: "... models can only simulate inundation in a small range.". Also, this is not entirely true, because in two-dimensional flood simulation the model computational limitations result from a combination of the simulation domain size and the spatial resolution of the data used.

- On the complexity of the model presented in this manuscript: if I understand well the maps presented in Figs 5 and 7, the number of catchments and the number of nodes is relatively small and should not be a problem for EPA SWMM model to handle. Perhaps I am missing something of the proposed method...

- On the spatial (elevation) data used: is a DEM of 30 m spatial resolution adequate to perform the proposed "detailed" analysis? what is the vertical and horizontal error of the DEM? Is the calculation of the average elevation and slope for the sub-catchments appropriate or does it create large errors? E.g. the slope calculation including the artificially added buildings to the DEM will increase the average slope for every sub-catchment (the slope at the edge of the buildings will be close to infinity!)

- there are a few questions about the equations presented (the equations are key to understand the proposed methodology): (1) in Page 7, Line 10, how was "r" defined?, (2) in "Step 1" (page 11, lines10-15), I do not see the difference between the two conditions... (3) Equation 6 seems to be wrong: how can variables of different units be subtracted ($h_i$ is a height (m) whereas $p$ seems to be a flow rate (m3/s))

- the tools used in some steps of the proposed methodology are not clear. For example, (1) "flow direction for each sub-catchment was calculated..." in Page 9, Line 6). But how? based on what tool? (2) how was catchment "width" (Page 10, lines 4) calculated? (3) how was the set of "optimal parameters" defined (Page 10, line 6)? How

was the calibration carried out?

- Results and conclusions: the results are somewhat expected, i.e. more rain -> higher flood depth. So, there is nothing novel here. In my opinion, the conclusion points reflect the problems mentioned above: – Point (1) it is not clear how EPA SWMM results are converted into flooding depth, – Point (2) Equation 6 is most likely wrong, – Point (3) English is very poor compromising the understanding of the text and the areas are not highlighted in the figures presenting the results – Point (4) it is obvious.

The quality of text can also be strongly improved, which may help the reader to follow the manuscript and understand the proposed methodology.

MINOR COMMENTS Page 1. 1st sentence of Abstract: "floods result (...) in recent years.". Recent years is in the Past, so the verb "result" needs to conjugated accordingly.

Page 1. "Schemed" scenario: what does "schemed" mean?

Page 1. Do metro stations have a pre-defined "drainage capacity"? how is it defined? do authors refer to existing pumping capacity? or something else? authors should explicitly define it.

Page 1. Lines 23-25: these sentences are not clear.

Page 2. Line 4: what exactly do authors mean by "geological" environment?

Page 2. Lines 11-12. "urban planning" is for the future and "prediction" is for the current urban layout. So, these sentences are not very coherent.

Page 3, line 14: what are "characteristics of the landform"?

Page 3, line 18: most of the hydrological studies and also urban flooding studies that I know take into account the catchment boundary as the boundary condition for the model. therefore, I disagree with the authors here. If the authors want to show their point, they should refer to previous studies including the appropriate references!

Page 4, 2nd paragraph: the 1st and last sentences of this paragraph do not match as they present opposite ideas.

Page 4, Line 24: what is "drainage station"?

Page 5, lines 6 and 7: is a reference needed to say where the Metropolitan area of Shanghai is?

Page 5, lines 9-13: the English quality of these sentences is very poor, compromising the understanding of the text.

Page 6, Line 6: Chicago design storm method does not "produce precipitation" but generates design hyetographs instead.

Page 7, line 4: who did the "documentary investigation"? who derived the IDF curves?

Page 8, line 25: what is "attention point"?

Page 9, Fig 2: where are the pumping stations presented in Fig 2? are they the same as drainage stations? flow direction arrows are not visible. How is sub-catchment drainage capacity calculated?

Page 13, line 10: why 2 hours for the simulation duration?

Page 17, line 1: how are "inundation ratios" calculated? this is not clear to me.

---

## Author Comment (AC7) · 29 Jun 2019

We have revised the manuscript hess-2019-28 according to the comments of the Referee#2. The revsed files are uploaded in the supparment. The supplements include the following documents: (1) revised manuscript with changes marked; (2) responses to the comments of the Referee#2ïijŻ (3) the figures in the revised manuscripts with pdf file; (4) English Language Certificate.

The authors would like to thank the constructive comments from the Referee#2, which are very helpful to guide the authors' revision on the manuscript.

Please also note the supplement to this comment:

[Figure]

https://www.hydrol-earth-syst-sci-discuss.net/hess-2019-28/hess-2019-28-AC7-supplement.zip

---

## Author Response (AR1)

Authors' Responses to the Comments of Referee #2

**hess-2019-28R2**
**Inundation analysis of metro systems using SWMM incorporated into GIS: a case study in Shanghai**

**Hai-Min Lyu, Shui-Long Shen, Jun Yang, and Zhen-Yu Yin**

The authors would like to thank the constructive comments from the reviewers, which are very helpful to guide the authors' revision on the manuscript. The revised parts are underlined in the revised manuscript. Authors' responses to the comments of reviewers are detailed as below, in which the paragraphs in normal fonts (in Cambria) are the original comments and the authors' responses are written in ***italic fonts (in Times New Roman)***.

**Comments**

Q1: -The topic of the manuscript is interesting and relevant. However, I have strong concerns about the proposed methodology because (perhaps) its poor description in the manuscript: - By including "Scenario-based..." in the title of the manuscript I was expecting something else than considering rainfall events of different return periods – this is classic in hydrology and I would not consider it a "scenario-based" analysis.

*Answer: Thanks for the reviewer's constructive comments. We have revised the title of the manuscript in the revised version as "Inundation analysis of metro systems using SWMM incorporated into GIS: a case study in Shanghai" to avoid confusion. We have added the description of the proposed method. The major contribution of this study is that the incorporation of the SWMM model into GIS model via a proposed water spreading algorithm. During the incorporation process, an algorithm is proposed to simulate the spreading process of rainwater on ground surface. The SWMM model is used to calculate the water volume of each subcatchment. The calculated water volume is adopted to perform the spreading process of each calculated grids of the study area in GIS model. We have revised the method to show how to incorporate SWMM model into GIS model. The revised section has been added from line 15 on page 7 to line 9 on page 8.*

**Line 15 on page 7 to line 9 on page 8:**
The SWMM model was incorporated into the GIS model to predict the inundation depth. The following phases must be performed during its incorporation:
(1) The investigated area was classified into different subcatchments in the GIS. Each subcatchment was provided with the corresponding geographical information (e.g. elevation, slope, area, and width). The information of each subcatchment was stored in the GIS database.
(2) The information of each subcatchment was exported from the GIS database and reproduced to produce a '.*inp*' document.

(3) The '*.inp*' document was integrated into the SWMM model to calculate the water volume of each subcatchment.

(4) The calculated water volume of each subcatchment was converted into the average water depth with the water volume and area of each subcatchment.

(5) Each subcatchment was divided into 20 m × 20 m grids in GIS. The study area includes 113810 grids. The information of each subcatchment and average water depth were extracted into the grids in GIS database.

(6) The grids with all information were applied to perform the spreading process with the proposed algorithm in GIS until the water level of each grid was stable. During spreading process, a spreading coefficient was used to move the runoff between neighbouring grids.

Finally, the water depth of each grid was exported to visualise the distribution of the inundation depth of the investigated area. The details of the proposed algorithm are presented in Section 3.2.

Q2: - The way the EPA SWMM model is connected to the GIS "model" is not clear. Also, EPA SWMM has two main parts: hydrology (lumped catchments) and hydraulics (pipes). How can EPA SWMM be used to estimate water depth on the terrain surface (see e.g. Page 4, line 8)? from EPA SWMM simulations one can obtain "flooding" results in each model node (representing e.g. a manhole), but it is a flow rate and not a depth (for the reasons indicated above).

*Answer: Thanks for the reviewer's comments. We have revised the manuscript to show how to connect the SWMM model to GIS model as shown in the response of Q1. In this study, the SWMM model cannot be directly used to estimate water depth on the terrain surface.*

*Yes, as pointed out by the reviewer that SWMM simulation can obtain the flow rate of each model node. The flow rate is considered as water volume of the model node. We incorporated the water volume of each model node into GIS to estimate the water depth of the surface. During the incorporation of SWMM into GIS, the water volume of model node in each subcatchment was calculated at first. Then, each subcatchment was meshed into grids with 20 m × 20 m in GIS. The study area includes 113810 grids. Each grid has its own geographical information (including elevation, slope, drainage capacity, land use type, and infiltration, etc.). Thirdly, it is to perform the spreading process of the calculated water volume. In this stage, we supposed that the calculated water volume of each subcatchment in SWMM model is uniformly distributed on the grids of the study area in GIS model at the beginning. After one cycle of iterative calculation, the water level will be not uniformly distributed and changed with the geographical information. The uniformed water depth is then adopted to perform the circulation of updating the water depth of each grids until the water level of each grind is stable. Finally, the water level of each grid is obtained to represent the water depth of the terrain surface. The approach of how to integrate SWMM into GIS is revised in the response of Q1.*

Q3: - The literature cited in the manuscript is rather old. For example, the authors cite the 2002 study from Horrit and Bates. More than 15 years have passed since this study was published and significant developments in terms of computational power have occurred. The authors should include more recent studies that might contradict their argument: "... models can only simulate inundation in a small range.". Also, this is not entirely true, because in two-dimensional flood simulation the model computational limitations result from a combination of the simulation domain size and the spatial resolution of the data used.

*Answer: Thanks for the reviewer's comments. We have deleted some previous references, and we have added the context to discuss the application of the SWMM model. The SWMM model is mainly applied to analyze inundation risk with small region, because the limitations of the SWMM model in 2D flood simulation result from the simulation domain size and the spatial resolution of the data used. We have added the revised section from line 18 on page 3 to line 3 on page 4.*

**Line 18 on page 3 to line 3 on page 4:**
However, the SWMM has been applied to only small regions of several square kilometres owing to its computational limitations in terms of simulation domain size and spatial resolution of data for 2D flood simulations (Wu et al., 2018; Chen et al., 2018; Kumar et al., 2019). For example, Zhu et al., (2016) applied the SWMM and a multi-index system to evaluate the inundation risks in southwest Guangzhou, China (area of 0.43 $km^2$). Feng et al. (2016) selected the SWMM as modelling platform to simulate the inundation risks for a campus in Salt Lake City, Utah, US (area of 0.11 $km^2$). Moreover, Wu et al. (2017) applied the SWMM in combination with LISFLOOD-FP to simulate urban inundations in Dongguan City, China (area of 2 $km^2$).

Q4: - On the complexity of the model presented in this manuscript: if I understand well the maps presented in Figs 5 and 7, the number of catchments and the number of nodes is relatively small and should not be a problem for EPA SWMM model to handle. Perhaps I am missing something of the proposed method...

*Answer: Thanks for the reviewer's comments. The Figs. 5 and 7 are obtained from GIS, which is not directly obtained from SWMM. The SWMM is just a tool to help to calculate the water volume of each subcatchment. Yes, the number of subcatchments and nodes are relatively small, but the study area is meshed into 113810 grids. The calculated water volume is adopted into GIS to perform the spreading process to update the water level of each grid, until the water level is stable. The water level of each grid is exported from GIS to represent the water depth of the terrain surface in Fig. 5. The water depth of the surface is then used to calculate the inundation depth of the metro station, which is presented in Fig. 7. We have illustrated the method of how to integrate SWMM into GIS. This section has been revised in the response of Q1.*

Q5: - On the spatial (elevation) data used: is a DEM of 30 m spatial resolution adequate to perform the proposed "detailed" analysis? what is the vertical and horizontal error of the DEM? Is the calculation of the average elevation and slope for the sub-catchments appropriate or does it create large errors? E.g. the slope calculation including the artificially added buildings to the DEM will increase the average slope for every subcatchment (the slope at the edge of the buildings will be close to infinity!)

*Answer: Thanks for the reviewer's comments. The original DEM data with 30 m resolution is reprocessed in GIS. During the reprocess of the elevation data, we haven't considered the vertical and horizontal error. We just considered the original DEM data and the distribution of buildings in the study area. The study area is classified into grids with 20 m × 20 m in GIS, and each grid can be given an elevation data based on the original DEM data. We extracted the original DEM data to each grid. The grids with building locations are modified to add the height of building. We overlaid the original DEM data and the distribution of the buildings with their corresponding heights. Of course, the surface slop will increase. The flood event is impossible to inundate the building. The area with the location of building will not inundated in flood event. Thus, the modification is reasonable and the data with 30 m resolution is enough to perform the rainfall spreading process with 20 m × 20 m grids in the study area. We have added discussions from line 5 to line 10 on page 7.*

**Lines 5-10 on page 7:**

During the reprocess of the elevation data, the original DEM data and building distribution with corresponding heights were overlaid. Furthermore, the investigated area was divided into grids with 20 m × 20 m in GIS. Each grid was provided with building distribution data and a DEM with spatial resolution of 30 m. The grids with the original DEM data were modified to include the building heights. Because locations with buildings are not inundated, the modification is reasonable. Furthermore, DEM data with 30 m resolution is sufficient for a range of 120 km$^2$.

Q6: - there are a few questions about the equations presented (the equations are key to understand the proposed methodology): (1) in Page 7, Line 10, how was "r" defined?, (2) in "Step 1" (page 11, lines10-15), I do not see the difference between the two conditions... (3) Equation 6 seems to be wrong: how can variables of different units be subtracted (hi is a height (m) whereas p seems to be a flow rate (m$^3$/s)).

*Answer: Thanks for the reviewer's helpful and detailed comments. We have answered the questions point by point.*

*(1) The parameter r is defined as the ratio of the time for the peak to the total event duration, is empirically fixed as 0.45 in Shanghai (Yin et al., 2016a). The parameter r is used to determine the location of the rainfall peak during the produce of the rainfall scenario. It has been revised from line 17 to line 19 on page 6.*

*(2) The step 1 is used to determine the location of the calculated grids in GIS. If the grid is surrounded by other 8 grids ($h_I + \Delta x = h_{Ij}$ or $h_I + \Delta y = h_{Ij}$), the spreading coefficient is determined as f = 1. If the grid located in boundary with less 8 surrounding grinds ($h_I + \Delta x = h_{Ij}$ and $h_I + \Delta y = h_{Ij}$), the spreading coefficient is determined as f = 0.569. We have revised this section from line 6 to line 10 on page 12.*

*(3) Thanks for the reviewer's detailed comments. The equation (6) has been revised. The meaning of the Eq. (6) is that, the remained rainwater minuses the height of the metro step and the drainage capacity of underground space, is used to judge whether the metro station will suffer from inundation. This section has been revised from line 10 on page 13 to line 5 on page 14.*

**Lines 17-19 on page 6:**
To consider temporal variations, the parameter *r* (the ratio of the time necessary to reach the peak to the total event duration) was empirically fixed to 0.45 (Yin et al., 2016a). The parameter *r* is used to determine the location of the rainfall peak during a rainfall scenario.

**Lines 6-10 on page 12:**
*Step 1*: The grid location (*GL*) and spreading coefficient (*f*) are determined (see Fig. 3a). Each grid $h_I$ is surrounded by $h_{Ij}$ grids ($j = 1, 2,…, 8$). If the grid is surrounded by eight grids ($h_I + \Delta x = h_{Ij}$ or $h_I + \Delta y = h_{Ij}$), the locations of grids $h_{Ij}$ are determined with $GL = 1$ and spreading coefficient $f = 1$. However, if the grid is located at a boundary and

surrounded by fewer than eight grids ($h_I + \Delta x = h_{Ij}$ and $h_I + \Delta y = h_{Ij}$), the locations of grids $h_{Ij}$ are determined with $GL = -1$ and spreading coefficient $f = 0.569$.

**Line 10 on page 13 to line 5 on page 14:**
The inundation depth around a metro station is used to evaluate the inundation risk of metro lines. Therefore, Eq. (6) is proposed:

$$h_{t(station)} = h_i - \frac{p}{A} - h_{0(station)} , \qquad (6)$$

where $h_{t(station)}$ is the inundation depth around the metro station, $\underline{h_i}$ the inundation depth on the ground surface after flooding events, $p$ the drainage capacity of the metro station, $\underline{A}$ the inundation area, and $h_{0(station)}$ the step height of the metro station (based on the design standards of metro systems, $h_{0(station)} = 0.2$ m). According to the code for metro design (GB 50157-2013), the drainage capacity of a metro station is determined according to the local IDF with a duration of 5 min for a 50-year rainfall intensity. In this study, the drainage capacity of the metro station was determined with Eq. (3) for a 50-year rainfall intensity and duration of 5 min. When $h_{t(station)} > 0$, the metro station becomes inundated.

Q7: - the tools used in some steps of the proposed methodology are not clear. For example, (1) "flow direction for each sub-catchment was calculated..." in Page 9, Line 6). But how? based on what tool? (2) how was catchment "width" (Page 10, lines 4) calculated? (3) how was the set of "optimal parameters" defined (Page 10, line 6)? How was the calibration carried out?

*Answer: Thanks for the reviewer's comments. We have added the related tools of the proposed methodology in the revised manuscript as the reviewer SC2 suggested.*

*(1) The flow direction of each subcatchment is determined from high to low in using the Hydrologic Analysis Tools in GIS. We supposed that the rainwater is flowed from the subcatchment with high elevation to the subcatchment with low elevation. The two adjacent subcatchments are connected by the flow direction.*

*(2) The width of each subcatchment can be calculated using the Spatial Analyst Tools in GIS.*

*(3) We have revised the section to define the parameters of the model. The calibration of the model using the comparison between the predicted results and the historic inundation locations. This section has been revised from line 10 on page 10 to line 9 on page 11, and line 15 to line 17 on page 14.*

**Line 10 on page 10 to line 9 on page 11:**
The impervious parameter was determined according to the land use types. The study area is located in the urban centre, where the land use has no big changes. The dense distribution of buildings leads to an impervious surface of more than 80% of the total surface. Owing to the existence of road pavements, subgrades, and many municipal pipelines under the roads, water infiltration through the road and subsurface is very low. Thus, roads can be considered impervious, and soil infiltration and evapotranspiration

have slight effects on the surface runoff concentration during short-term flash flooding during rainstorms. The soil infiltration mainly depends on green land (e.g. lawns, flower beds, and groves) and in the water bodies within the study area. The geotechnical information in Shanghai is as follows: The groundwater table is higher than 2 m below the ground surface. The soil type at a depth of 2 m is mixed soil with sand (5%), silt (55%), and clay (40%) according to the Shanghai Geotechnical Investigation Code (DGJ08-37-2012). The sand content is 15% at the surface. Thus, the soil has a hydraulic conductivity of $2 \times 10^{-5}$ m/s, which is 72 mm/h. At the bottom of the water body, the soil has more clay (>50%) and less sand (<5%) with a hydraulic conductivity of $2 \times 10^{-7}$ m/s, which is 0.72 mm/h (Shen et al., 2015). Based on the SWMM handbook, the maximal infiltration rate was set to 72 mm/h to reflect the characteristics of the green land, whereas the minimal value (0.72 mm/h) reflects the characteristics of the water body because the underlying soil is saturated clay. In addition, the blocking effect of the buildings has a significant influence on the surface runoff generation and concentration. Therefore, the heights of existing buildings were extracted to modify the elevation of the calculated grids.

**Lines 15-17 on page 14:**
The calibration of the proposed model is based on a comparison between the predicted results and historic inundation locations.

- Results and conclusions: the results are somewhat expected, i.e. more rain -> higher flood depth. So, there is nothing novel here. In my opinion, the conclusion points reflect the problems mentioned above: – Point (1) it is not clear how EPA SWMM results are converted into flooding depth, – Point (2) Equation 6 is most likely wrong, – Point (3) English is very poor compromising the understanding of the text and the areas are not highlighted in the figures presenting the results – Point (4) it is obvious.

*Answer: Thanks for the reviewer's comments.*

*Point (1): We have revised the approach of how to incorporate SWMM into GIS to convert into flooding depth. This section has been revised in the response of Q1.*

*Point (2): Thanks for the reviewer's constructive comments. The equation (6) has been revised.*

*Point (3): The revised manuscript has been proofed by the English Language Service.*

*Point (4): The conclusion (4) has been deleted. We have added the conclusion of how to integrate SWMM model into GIS model in conclusion (2). This section has been revised from line 16 to line 21 on page 25.*

**Lines 16-21 on page 25:**
**Conclusions:**
(2) The study area was classified into subcatchments, and their corresponding information was stored in the GIS database. The information of each subcatchment was exported and input in the SWMM model to calculate the water volume of each

subcatchment. Moreover, each subcatchment was meshed into grids. The calculated water volume in the SWMM model was adopted to update the water level of each grid in GIS with the proposed algorithm. Finally, the stable water level of each grid in GIS was used to determine the inundation depth.

Q8: The quality of text can also be strongly improved, which may help the reader to follow the manuscript and understand the proposed methodology.
*Answer: Thanks for the reviewer's comments. We have revised the methodology to make it more understandable and the English has been proofed by the native speaker of Elsevier Language Service.*

Q9: MINOR COMMENTS
1) Page 1. 1st sentence of Abstract: "floods result (...) in recent years.". Recent years is in the Past, so the verb "result" needs to conjugated accordingly.
*Answer: We have revised the "result" into "have resulted".*

2) Page 1. "Schemed" scenario: what does "schemed" mean?
*Answer: We have revised the "schemed" into "designed".*

3) Page 1. Do metro stations have a pre-defined "drainage capacity"? how is it defined? do authors refer to existing pumping capacity? or something else? authors should explicitly define it.
*Answer: Thanks for the reviewer's valuable comments. Based on the Code for design of metro (GB 50157-2013), the drainage capacity of metro station is determined according to the local intensity-duration-frequency (IDF) within the rainfall duration of 5 minutes. In this study, the drainage capacity is calculated using Eq. (3) with the rainfall duration of 5 minutes. We have added the definition of the drainage capacity of metro station from line 2 to line 5 on page 14. The drainage capacity of the existing drainage station is obtained from literatures (Yin et al., 2016b), which has been presented in Fig. 2. The revision to illustrate the drainage capacity of the drainage station has been added from line 10 to line 11 on page 9.*

**Lines 2-5 on page 14:**
According to the code for metro design (GB 50157-2013), the drainage capacity of a metro station is determined according to the local IDF with a duration of 5 min for a 50-year rainfall intensity. In this study, the drainage capacity of the metro station was determined with Eq. (3) for a 50-year rainfall intensity and duration of 5 min.

**Lines 10-11 on page 9:**
The drainage capacity of each drainage station was obtained from the existing publication (Yin et al., 2016b).

[Figure]

**Figure 2:** Calculated subcatchments and grids in SWMM and GIS: (a) drainage capacity and flow direction of each subcatchment; (b) calculated grid of each subcatchment

**Lines 10-11 on page 9:**
The drainage capacity of each drainage station was obtained from the existing publication (Yin et al., 2016b).

(This is a computer generated advice and does not require any signature)

---

## Author Response (AR2)

**Authors' Responses to the Comments of Report #2**

**hess-2019-28R3**
**Inundation analysis of metro systems using SWMM incorporated into GIS: a case study in Shanghai**

**Hai-Min Lyu, Shui-Long Shen, Jun Yang, and Zhen-Yu Yin**

The authors would like to thank the constructive comments from the reviewers, which are very helpful to guide the authors' revision on the manuscript. The revised parts are underlined in the revised manuscript. Authors' responses to the comments of reviewers are detailed as below, in which the paragraphs in normal fonts (in Cambria) are the original comments and the authors' responses are written in *italic fonts (in Times New Roman)*.

**Editor Comments**

Please, address comments by Referee #2, who revised your manuscript twice and still is points at numerous necessary improvements.
*Answer: Thanks for the Editor give us the chance to revise our manuscript and sincere thanks also for the second anonymous referee, who carefully revised our manuscript to improve the quality. We have revised our manuscript according to the reviewer's comments point by point.*

**Comments**

Despite the changes carried out by the authors to the previous version of the manuscript, there are still a few points that I consider worth revising:
Q1- Line 18 (page 3) to line 3 (page 4): SWMM is not a 2D model. So, how can SWMM be limited by the 2D domain size and spatial res?
*Answer: Thanks for the reviewer's comments. Yes, SWMM is a 1D model, which can be adopted to calculate the water flowing velocity in pipeline and water volume of each subcatchment. However, SWMM cannot be used to determine the surface water flowing and the redistribution of rainwater. Thus, in this study, we adopt SWMM to calculate water volume of each subcatchment and then proposed the algorithm to determine the spreading of the calculated water volume. With this spreading algorithm, we incorporated SWMM into GIS. We have revised this sentence from line 23 to line 24 in page 3.*

**Line 23-24 in page 3:**
However, the SWMM cannot be used to determine the surface water flows.

Q2- In the answer to Q4: Is "113810" a large number of cells? I believe there are other studies that encompass a larger number of grid cells to simulate 2D surface flow.

*Answer: Thanks for the reviewer's comments. Yes, I agree with the reviewer that there are other studies, which include a larger number of cells to simulate 2D surface flow. Here, we would like to demonstrate that the simulation model with 113810 grid cells is enough to simulate the area in our case. The water depth (Fig. 5) is obtained from the calculated grid rather than from the subcatchment. The number of subcatchment is relatively small, but the study area is meshed into 113810 grids. The water depth is obtained from the calculated grids by using the proposed spreading algorithm in GIS.*

Q3- In Answer to Q5: How is the resolution converted from 30m to 20m? what type of interpolation? Also, what level of accuracy is expected to have when adding the buildings to a raster of 20m spatial res??? More detailed is required and well as a discussion on this.

*Answer: Thanks for the reviewer's comments. The elevation of each grid was extracted from the DEM data with 30 resolution using the tool 'from point to grid' in GIS. The theory of this process is linear interpolation. Since the Shanghai is a flat region and the difference of elevation is only 2 to 3 m within a range of 25 km. Thus, the accuracy of linear interpolation will be very small, e.g., slope is less than 0.01%, the error from interpolation will be also less than 0.01% both horizontally and vertically. By using this process, each grid was given an elevation. Then, the elevation of each grid was modified by the distribution of building. The grids with building locations are modified to add the height of building. We overlaid the original DEM data and the distribution of the buildings with their corresponding heights. Of course, the surface slop will increase. The flood event is impossible to inundate the building. The area with the location of building will not inundated in flood event. Thus, the modification is reasonable and the data with 30 m resolution is enough to perform the rainfall spreading process with 20 m × 20 m grids in the study area. Since the study area is classified into grids with 20 m × 20 m in GIS, the calculated water depth is within 400 m². In other words, the surface water depth can be predicted in the accuracy of the area within 400 m². We have revised section from line 5 to line 11 in page 7.*

**Line 17 in page 7 to line 2 in page 8:**

During the reprocess of elevation data, the investigated area was divided into grids with 20 m × 20 m in GIS. Since the original elevation of each grid was extracted from the DEM data with spatial resolution of 30 m, the linear interpolation was conducted to convert it into 20 m size. Shanghai is a flat region with an elevation difference of only 2 to 3 m within a range of 25 km. Thus, the accuracy of linear interpolation is enough. Each grid was provided with building distribution data and a DEM. The grids with the original elevation were modified to include the building heights. Because the locations with buildings are not inundated, the modification is reasonable. Since the study area is

classified into grids with 20 m × 20 m, the obtained distribution of water depth is within the area of 400 m². Therefore, the proposed method can achieve an accuracy of the inundation distribution within 400 m².

Q4- In point (2) of the conclusions: "… the calculated water volume in the SWMM…" refers to flooding volume, right? But how is water spread in the catchment grid cells?

*Answer: Thanks for the reviewer's comments. Yes, the calculated water volume is flood volume of each subcatchment. The calculated water volume is used to perform the spreading process using the proposed algorithm. In this stage, it is supposed that the calculated water volume of each subcatchment is uniformly distributed on the grids of the study in GIS model at the beginning. After one cycle of iterative calculation, the water level will be not uniformly distributed and changed with the geographical information. The uniformed water depth is then adopted to perform the circulation of updating the water depth of each grids until the water level of each grind is stable. Finally, the water level of each grid is used to reflect the surface water depth. The detailed process about how to incorporate SWMM into GIS has been revised from line 6 to line 24 in page 8. The water volume is adopted to spread into each grid using interactive calculation with the proposed algorithm. We have revised this section from line 9 to line 10 in page 26.*

**Line 9-10 in page 26:**
The calculated water volume was adopted to update the water level of each grid using iterative calculation with the proposed algorithm.

Q5- Lines 12-13 (page 3): difficult to model? most GIS software do it automatically based on DEMs-based algorithms.

*Answer: This sentence is used to say that the numerical model is difficult to model the characteristics of landform. Yes, most GIS software can do it automatically based on DEMs-based algorithms. This study adopted the GIS software to reflect the characteristics of landform.*

Q6- In Answer to Q8: The authors should not "delete" parts of the text to avoid confusion. They should instead clarify the steps and concepts used in the text.

*Answer: Thanks for the reviewer's comments. We deleted the sentence "*Moreover, the existing numerical studies cannot identify the boundary, resulting in a large error because the boundary is in extreme vicinity of the area centre.*". This original sentence did not express the correct meaning. Thus, we revised the context and added references to support our points. This section has been revised from line 16 to line 19 in page 3.*

**Line 16-19 in page 3:**
Many of the existing methods can only simulate inundation for small ranges (Naulin et al., 2013; Wu et al., 2018). Therefore, a new tool (e.g., the GIS technique) is required

to consider variations in topographical elevations. Moreover, an integrated method is required to simulate regional-scale flooding.

Q7- In Answer to Q10 ("… instead we use the drainage capacity of the drainage system…"): This is a very strong assumption! replace the drainage system by the metro stations "hyd. capacity"? there are other methods, e.g. reduce a few mm from the rainfall.

*Answer: We haven't used the metro station to replace the drainage system. Excuse our poor English to cause the misunderstanding. Since the distribution of the drainage network is complex in the study region with 120 km² so that we haven't considered the effects of the drainage network directly. Instead, we use the drainage capacity of the pumping station to reduce the calculated water quantity to indirectly reflect the effects of the drainage network. The calculated drained water volume was already deduced in SWMM model. The distribution of the pumping station is shown in Fig. 2. Moreover, there is a mistake for the term "drainage station", it should be "Pumping station".*

[Figure]

**Figure 2:** Calculated subcatchments and grids in SWMM and GIS: (a) drainage capacity and flow direction of each subcatchment; (b) calculated grid of each subcatchment

Q8- Line 3 (page 6): I think the authors mean "generate rainfall hyetographs" instead of "produce rainfall processes"
*Answer: Thanks for the reviewer's careful check. We have revised it as "generate rainfall hyetographs" in line 6 (page 6).*

Q9- Lines 1-2 (Page 24): "practical" interest??? the duration may be related to the catchment time of concentration… flash floods? … I think the authors should describe this in more detail.

*Answer: Thanks for the reviewer's comments. In realty, people paid more attention to the short-term rainstorm. Since a short-term rainstorm is more possible to induce flash floods in urban area. Therefore, in this study, we simulate a rainfall duration with 2 h under the rainfall intensity of 50-year, 100-year and 500-year. We have revised this section in more detail from line 13 to line 15 in page 24.*

**Line 13-15 in page 24:**
In reality, a short-term rainstorm is easy to induce floods in urban area. The existing researches paid more attention to flash floods induced by short-term rainstorm within 2 h or 3 h (Yin et al., 2016a, b; Wu et al., 2017). Therefore, in this study, a rainfall duration of 2 h was selected to simulate.

**Authors' Responses to the Comments of Report #1**

**hess-2019-28R3**
**Inundation analysis of metro systems using SWMM incorporated into GIS: a case study in Shanghai**

**Hai-Min Lyu, Shui-Long Shen, Jun Yang, and Zhen-Yu Yin**

The authors would like to thank the constructive comments from the reviewers, which are very helpful to guide the authors' revision on the manuscript. The revised parts are underlined in the revised manuscript. Authors' responses to the comments of reviewers are detailed as below, in which the paragraphs in normal fonts (in Cambria) are the original comments and the authors' responses are written in ***italic fonts (in Times New Roman)***.

**Comments**

This manuscript aims to evaluate the flood risk of metro systems under different rainfall intensities of 50-year, 100-year and 500-year. The incorporation of SWMM and GIS is used to simulate the water depth. To achieve this goal, an algorithm to simulate the surface flow is proposed. The topic is interesting and within the scope of the journal of hydrology and earth system science. I carefully read the revised manuscript as well as the interactive comments of the discussers and reviewers and the responses of the authors. The proposed comments are suggestive and helpful to improve the quality of the manuscript, and the authors have answered the questions carefully. Therefore, I suggest the current version of the revised manuscript can be accepted. In addition, one question is launched for the authors to consider in their future research activities: what is the effect of the land subsidence on the flood risk and how to simulate this effect?

*Answer: Thanks for the reviewer's constructive comments. This is a useful suggestion for the future studies. Yes, in the present study we did not considered the effect of land subsidence on flooding directly. Land subsidence is a slow process of loss of elevation and this is a dynamic process so that further detailed study (calculation algorithm) is required to consider the interaction between elevation loss and flooding.*